# Mitochondrial Properties in Skeletal Muscle Fiber

**DOI:** 10.3390/cells12172183

**Published:** 2023-08-30

**Authors:** Han Dong, Shih-Yin Tsai

**Affiliations:** 1Department of Physiology, Yong Loo Lin School of Medicine, National University of Singapore, Singapore 117593, Singapore; e0427779@u.nus.edu; 2Healthy Longevity Translational Research Programme, Yong Loo Lin School of Medicine, National University of Singapore, Singapore 117456, Singapore

**Keywords:** mitochondria, skeletal muscle physiology

## Abstract

Mitochondria are the primary source of energy production and are implicated in a wide range of biological processes in most eukaryotic cells. Skeletal muscle heavily relies on mitochondria for energy supplements. In addition to being a powerhouse, mitochondria evoke many functions in skeletal muscle, including regulating calcium and reactive oxygen species levels. A healthy mitochondria population is necessary for the preservation of skeletal muscle homeostasis, while mitochondria dysregulation is linked to numerous myopathies. In this review, we summarize the recent studies on mitochondria function and quality control in skeletal muscle, focusing mainly on in vivo studies of rodents and human subjects. With an emphasis on the interplay between mitochondrial functions concerning the muscle fiber type-specific phenotypes, we also discuss the effect of aging and exercise on the remodeling of skeletal muscle and mitochondria properties.

## 1. Introduction

Mitochondria are double-membrane-bounded organelles shared by most eukaryotic cells, which are derived from the mitochondrion-bacteria and have acted as a symbiotic partner of nuclear–cytosolic organisms for over 2 billion years [1]. Although most mitochondrial genes (>1100) are encoded by nuclear DNA (nDNA), mitochondria itself reserves 37 genes in its genome (mtDNA). Among these 37 genes, 13 encode the protein for mitochondria oxidative phosphorylation enzyme, and 25 encode the tRNA and rRNA for mitochondria protein translation [2]. The main function of mitochondria is to generate energy. Mitochondria generates energy in the form of adenosine triphosphate (ATP) from energy-enriched molecules such as pyruvate, fatty acids, and amino acids via oxidative phosphorylation [3]. Electrons generated from oxidations of energy-enriched molecules are transferred via nicotinamide adenine dinucleotide hydrogen (NADH) to complex I (NADH ubiquinone oxidoreductase) or flavin adenine dinucleotide (FADH_2_) to complex II (succinate dehydrogenase), then transported to coenzyme Q. Coenzyme Q then delivers electrons generated from complex I or II via complex III (cytochrome bc1 complex) to cytochrome c and then to complex IV (cytochrome c oxidase), where oxygen is reduced to water. Finally, coupling with electron generation, the protons (H^+^) are pumped to the intermembrane space from complex I, III, and IV for ATP production in complex V (ATP synthase). Since complex II does not actively pump proton to the intermembrane space, it contributes less ATP production. In addition to ATP generation, mitochondria have other functions, including the production of reactive oxygen species (ROS) and the regulation of cellular calcium homeostasis. Mitochondrial ROS is the side product of the incomplete mitochondrial oxidative phosphorylation process from the electron leakage predominately in complexes I and III. Excess ROS damages cells by oxidation of nucleic acids, proteins, and lipids. Yet, the growing evidence reveals that ROS acts as a secondary messenger that participates in a wide range of cell signaling to stimulate cell proliferation, differentiation, death, etc. [4,5,6,7]. In collaboration with the extracellular matrix and the endoplasmic reticulum (ER), mitochondria also serve as a transient calcium sink to regulate cytosolic calcium levels. Mitochondrial calcium homeostasis has been implicated in controlling ATP production and cell death evoked by mitochondria [8,9,10]. Mitochondrial calcium overload perturbs mitochondrial membrane potential and mitochondrial dysfunction, unavoidably leading to cell death. Due to the critical role of mitochondria in regulating energy metabolism and diverse cellular functions, a good population of healthy mitochondria is necessary for cell survival, especially for high energy-demanded tissues such as skeletal muscle. In skeletal muscle, mitochondria are primarily distributed within the subsarcolemmal area (grouped beneath the plasma membrane) and intermyofibrillar area (nested between parallel myofiber). The intermyofibrillar mitochondria can be further separated into two subpopulations: one resides at the I-band, which contains only the actin filament of muscle fiber, tethering the sarcoplasmic reticulum network, and the other is located at the A-band, which contains both actin and myosin filaments across the myofiber, closely to the capillaries. 

In humans, skeletal muscle comprises 40% to 50% of the body mass and accounts for about 30% of the basal energy expenditure [11]. Compared with other muscle tissues, such as cardiac and smooth muscle, skeletal muscle cells fuse together to form elongated muscle fibers, which have multiple nuclei that are peripheral to the fiber. The skeletal muscle fibers are highly heterogeneous, reflecting their various contractile properties (slow or fast) and metabolic adaptations (oxidative or glycolytic). The control of skeletal muscle is voluntary by motor neurons to generate force and locomotion. The coordination of differences in nerve impulse transmission, membrane excitability, excitation–contraction coupling calcium flux between sarcoplasmic reticulum and cytosol, and ATP hydrolysis rate of myosin ATPase generates a variety of movements in our daily life. Most, if not all, of the cellular actions controlling movement are highly dependent on mitochondrial activities. It was not surprising that the common feature of mitochondrial diseases is muscle dysfunction. Moreover, mitochondrial damage and impairment are thought to be the driving forces that trigger the process of muscle aging. In contrast, aerobic exercise has been shown to improve skeletal muscle performance by increasing mitochondrial biogenesis and turnover, promoting intrinsic mitochondrial functions [12,13]. To fulfill their unique contractile properties, skeletal muscle fibers differ in mitochondrial activities, dynamics, and quality control. In this review, we summarize the recent findings on mitochondrial specialization in different skeletal muscle fibers and their physiologic relation to skeletal muscle health. Readers who are interested in mitochondria in the heart and other systems can refer to the previous excellent reviews [14,15].

## 2. Mitochondria in Different Types of Skeletal Muscle Fibers

Skeletal muscle is composed of a mixed collection of fiber types. Each muscle fiber contains repeating units of actin and myosin filaments, named sarcomeres, arranged in a stacked pattern whose cyclical interactions produce muscle movement. The power output from a muscle fiber shortening relies on the intrinsic ATP hydrolysis activity of the myosin heavy chain (MyHC). In general, muscle contraction initiates with an action potential traveling along α-motor, which triggers a release of acetylcholine from the axon terminal in the neuromuscular junction (NMJ). Acetylcholine then binds to acetylcholine receptors (AChR) located in the center of the myofiber and initiates a subsequent depolarization of the muscle fibers, causing calcium to leave the sarcoplasmic reticulum and bind to troponin on the actin molecule. The binding of calcium and troponin displaces tropomyosin from actin, allowing interaction between actin and myosin to generate a muscle contraction. Once the chemical signal from the motor neuron ends, the calcium gate on the sarcoplasmic reticulum closes. The calcium will be pumped back to the sarcoplasmic reticulum via ATP-driven pumps, and the binding between myosin and actin will then re-inhibit to let the muscle relax [16].

Based on the predominant expression of sarcomeric MyHC isoforms, adult mammalian skeletal muscle can be classified into four subtypes: type I, type IIa, type IIb, and type IIx, which highly express MyHC-I, MyHC-IIa, MyHC-IIb, and MyHC-IIx, respectively. MyHC-IIb is characterized by a relatively faster shortening velocity and higher ATP hydrolysis rate, followed by MyHC-IIx, MyHC-IIa, and MyHC-I lowest. There are also hybridized fibers that express more than one kind of MyHC at high levels to fulfill the requirement for diverse movements. In general, type II fibers are classified as “fast-twitched fibers”, while type I fibers are “slow-twitched fibers” based on the ATP hydrolysis rate of their corresponding MyHC expression [17,18]. Often, the MyHC composition of muscle fiber is related to their metabolic properties. Even though glycolysis yields less ATP than oxidative respiration, fast muscle fibers rely on the rapid ATP generation from glycolysis. In comparison, slow-twitch muscle fibers take advantage of sustained ATP production from oxidative respiration to support their long-duration activities. Histochemical staining for α-glycerophosphate dehydrogenase (GPD) activity indicative of glycolytic metabolism is higher in MyHC-IIb and MyHC-IIx rich fibers, intermediate in MyHC-IIa, and lowest in MyHC-I. Reversely, MyHC-IIa-rich fibers display the highest succinate dehydrogenase (SDH) activity indicative of oxidative metabolism, followed by MyHC-I > IIx > IIb in rat plantaris. Similar to rat plantaris, human vastus lateralis has higher GPD activity in MyHC-IIx fibers, intermediate activity in MyHC-IIa, and lower activity in MyHC-I, in which no MyHC-IIb expressed fibers were identified in adult human limb muscles, whereas the relative SDH activity is reversed. By combining the classifications of contractile and metabolic properties, muscle fibers are further classified into three subgroups: slow-oxidative fibers (type I), fast-oxidative fibers (type IIa), and fast-glycolytic fibers (type IIx and type IIb) [19].

Muscle fiber diversity is generally thought to be determined by its corresponding α-motor neuron for controlling a variety of force and motor tasks. Three subgroups of α-motor neurons are characterized by their excitability and firing pattern to control muscle contraction. Based on impulse activity, long-lasting trains, and frequency of firing measured by continuous electromyography recording implanted in the rat muscles, α-motor neurons coupled with their controlled muscle fiber types are classified as fast-fatigable (innervated type IIb or IIx fibers), fatigue-resistant (innervated type IIa fibers), and slow (innervated type I fibers) motor units. The impulse activity per 24 hrs and train duration is higher in slow α-motor neurons, intermediate in fatigue-resistant α-motor neurons, and lowest in fast-fatigable α-motor neurons. Correspondingly, type I fibers are often used to maintain posture and stabilize bone and joints, and type IIa fibers are explicated long-lasting and repetitive movements, such as respiration. Yet, fast-fatigable α-motor neurons have a relatively higher firing frequency, followed by fatigue-resistant and slow α-motor neurons. Thus, type IIx and IIb fibers are fit for rapid and robust movements, such as sprinting and weightlifting [20]. 

Muscle is a highly plastic tissue that can alter its size, metabolic properties, fiber type proportion, and other parameters during its life course. The plasticity of muscle fiber adapting to demands from various motor tasks highly relies on the modulation of mitochondria activities, such as ATP generation and calcium uptake capacity. In this section, we summarize the recent findings on profiling mitochondria properties in different types of skeletal muscle fibers, as well as their physiological relevance to muscle performance. Many other excellent reviews have focused on various aspects of motor neuron and neuromuscular junctions in regulating skeletal muscle functions, and the readers are thus referred to them [21,22]. 

### 2.1. Mitochondria Content and Oxidative Activity in Different Muscle Fiber Types

Mitochondria content and oxidative activity are closely related to the ATP demand of the muscle. Generally, type I and type IIa fibers contain more mitochondria compared to type IIx and IIb fibers, which positively correlate with increased oxidative activity measured by SDH histochemical assay as described above. Diverse muscle types have different levels of mitochondria content, which is controlled by transcription factors governing mitochondria biogenesis [23,24]. Peroxisome proliferator-activated receptor γ ( PPARγ) coactivator 1α (Pgc1α) is the most well-studied co-transcription factor, regarded as the core regulator of mitochondria biogenesis on the transcription level. In skeletal muscle, Pgc1α interacts with transcription factors: nuclear respiratory factor 1/2 (NRF1/2), whose target genes are mainly involved in the electron transport chain (ETC), and estrogen-related receptor α (ERRα), whose target genes are involved in the nuclear-encoded mitochondrial genes, as well as mitochondrial transcription factor A (TFAM), which regulates the replication of mtDNA and transcription of mitochondria-encoded genes [25]. Genetic modifications of these transcription factors have been demonstrated to affect fiber-type composition in mice. 

In transgenic mice with muscle-specific overexpression of *Pgc1α*, the mitochondrial oxidative phosphorylation system (OXPHOS) enzyme activities, such as SDH and cytochrome c oxidase (COX), were enhanced and correlated with an increase in the fiber type I and IIa (oxidative fiber) in muscle [26,27,28]. *Pgc1α* transgenic mice have an augmented mitochondrial count, and the genes related to OXPHOS and tricarboxylic acid (TCA) cycle are also increased in their muscle as a result. In contrast, mice with muscle-specific *Pgc1α* knockout by deleting floxed *Pgc1α* alleles using Cre recombinase under the control of the myogenin promoter and *Mef2c* enhancer (Myo-Cre) exhibit a muscle fiber type transition from oxidative fiber (type I and IIa) to glycolytic fiber (type IIb and IIx), as shown in the classification of muscle fiber type staining [27,29]. Targeted deletion of *Pgc1α* in muscles has weakened muscle strength and augmented muscle damage after strenuous treadmill exercise. 

Muscle fiber-specific single knockout of *Errγ* or *Errα*, two highly expressing ERR isoforms in skeletal muscle, using Cre recombinase driven by the human alpha-skeletal actin promoter (HSA-Cre) shows no significant impact on skeletal muscle architecture or functionality. Yet, double knockout of *Errγ/α* in skeletal muscle fiber (*Errγ/α*mKO) reduces the endurance running performance, as shown by shorter running distance and time on the treadmill, as well as the voluntary wheel [30,31]. RNA-seq analysis reveals that double knockout of *Errγ/α* reduces the expression level of genes involved in mitochondrial oxidative metabolism programs, which further confirms that ERRs are the main transcription factors that regulate mitochondria biogenesis. 

Moreover, loss of muscle Errγ/α lowers the vascular supply analyzed by immunostaining of CD31, an endothelial marker of capillary, in tibialis anterior (TA) muscle, and enhances muscle fatigue measured in the plantar flexion fatigue analysis. Genes involved in mitochondrial biogenesis and fatty acid oxidation were downregulated in *Errγ/α*-deficient gastrocnemius (GAS) muscle. Enzymatic activity of NADH and SDH for the mitochondrial complex I and II, respectively, are reduced in *Errγ*/α-deficient TA muscle. Correspondingly, mitochondrial respiration analyzed by seahorse essay was also lower in single muscle fiber isolated from the flexor digitorum brevis (FDB) muscle of *Errγ/α*mKO mice. Electron microscope results show that the size and density of mitochondria are significantly reduced in both subsarcolemmal and intermyofibrillar regions of TA muscle. Despite the reduced features of oxidative fiber, the characterization of muscle fiber type in TA muscle revealed an unexpected increase in a higher proportion of type IIa (oxidative fiber), lower type IIx (oxidative/glycolytic fiber), and comparable type IIb (glycolytic fiber) in *Errγ/α*mKO mice [31]. Conversely, overexpression of *Errγ* in skeletal muscle has been demonstrated to enhance oxidative capacity, as indicated by augmented endurance exercise capacity and higher mitochondria enzyme activity [32]. Converting glycolytic into oxidative fiber transition was evident in *Errγ* transgenic mouse muscle [33,34]. Overexpression of *Errγ* promotes fiber type transformation and is accompanied by the induction of angiogenesis-related genes, another feature of oxidative fiber [34]. Similar phenotypes of promoting muscle angiogenesis have also been reported in mice with overexpression of *Errα* in skeletal muscle [35].

Another transcription factor interacting with Pgc1α, Nrf2, activates not only gene expression involved in the ETC but also numerous antioxidant and detoxification enzymes in response to oxidative stress. Nrf2 was negatively regulated by the Kelch ECH-associating protein 1 (Keap1) complex. Under basal conditions, Nrf2 was sequestered in the cytoplasm by Keap1, which subjected Nrf2 to a ubiquitination–proteasome system for degradation. Thus, interruption of the binding between Nrf2 and Keap1 stabilizes and activates Nrf2. Mice with *Nrf2* or *Keap1* flox allele were crossed with transgenic mice expressing HSA-CRE-ER, where Cre recombinase is driven by human skeletal actin promoter and activated in the presence of tamoxifen; mice with tamoxifen-induced deletion of *Nrf2* or *Keap1* in adult skeletal muscle were generated. Tamoxifen was administered via intraperitoneal injection (2 mg/day for 5 consecutive days) in 4-month-old mice. Twenty weeks post-injection, 9-month-old *Nrf2* muscle knockout mice exhibited significantly reduced maximal running speed, distance, and duration on treadmill running tests and lower in situ muscle contraction evoked by electrical stimulation. Correspondingly, mitochondrial biogenesis and respiratory were reduced in *Nrf2*-deficient myofiber. In contrast, 9-month-old *Keap1* muscle knockout mice, which increase the Nrf2 protein level in skeletal muscle, have enhanced exercise capacity and maximal force generation, opposite to *Nrf2* muscle knockout mice [36].

Oxidative fibers have higher mitochondria volume density than glycolytic fibers, with greater mitochondria complex protein expression and mtDNA copy number [37,38]. As for the mitochondria content, mitochondria isolated from red and white porcine skeletal muscle using Percoll gradients showed no difference in mitochondria complex protein composition via a two-dimensional differential in gel electrophoresis between these two muscles [39]. Yet, respiratory activity in isolated mitochondria yielded a greater oxidative capacity when fatty acids (palmitoyl-carnitine and malate) acted as a substrate in porcine vastus intermedius muscles (70% oxidative fibers) than in gracilis muscles (70% glycolytic fibers). There is no difference in respiratory activities from other substrates, including carbohydrates (pyruvate and malate) or protein (glutamate and malate) between these two muscles. A similar result has been reported in isolated mitochondria from rat soleus muscles (oxidative fibers), which have greater respiratory activity on fatty acids compared to the extensor digitorum longus (EDL) muscles (glycolytic fibers) [40,41]. Single muscle fiber proteomics studies also confirm fiber-type differences in mitochondrial substrate oxidation. By proteomic analysis, the relative abundance of MyHC isoforms correlates with variations among mitochondrial proteomes in skeletal muscles from 3-month-old mice [42]. Mitochondrial enzymes in the mitochondrial matrix responsible for fatty acid oxidation are most prevalent in Myosin heavy chain 7 (Myh7)-enriched fibers (type I, slow-oxidative), and mitochondrial OXPHOS proteins are highly presented in Myh2-enriched fibers (type IIa, fast-oxidative). Proteins related to pyruvate metabolism and the TCA cycle are most abundant in Myh1-enriched fibers (type IIX, fast-glycolytic). The highest levels of GPD2, the mitochondrial component of the glycerophosphate shuttle, are present in Myh4-enriched fibers (type IIb, fast-glycolytic). The fiber type-specific profiles of mitochondrial proteomes are also present in humans. While humans did not have Myh4-enriched fibers, mitochondrial proteins involved in beta-oxidation, OXPHOS, and TCA cycles are greatly expressed in Myh7-enriched fibers. In contrast, mitochondrial enzymes involved in glycerophosphate shuttle are especially abundant in Myh1-enriched fibers, and Myh2-enriched fibers have intermediated expression for them [43]. 

As described above, the unique metabolic property of each muscle fiber type is thought to match the metabolic demand from different compositions of MyHC isoforms. Muscle-specific *Pgc1α* overexpression results in profound increases in oxidative properties in the skeletal muscle of transgenic mice. The *Pgc1α* transgenic mice not only have a greater endurance running capacity [26,27], but they also have a higher level of fatty acid oxidation in isolated mitochondria from their hind limbs [44], as well as in the ex vivo skeletal muscle fiber [45] as another indicator of enhanced oxidative properties of skeletal muscle. It is postulated that increased fatty acid oxidation can enhance endurance exercise performance. Peroxisome proliferator-activated receptor β (Pparβ, also known as Pparδ) is the major transcriptional regulator of fatty acid metabolism in skeletal muscle. Transgenic over-expression of *Pparβ* in skeletal muscle promotes fatty acid catabolism and endurance exercise performance [46,47]. Mitochondrial activities and contents are also increasing in *Pparβ* transgenic mice. Reversely, myofiber-specific ablation of *Pparβ* using HAS-Cre reduces fatty acid catabolism [48] without affecting fiber type specification in skeletal muscle [49]. Moreover, the deletion of *Pparβ* abolishes exercise-induced metabolic shift from glucose to fatty acid, which in turn weakens the improved endurance capacity upon exercise training [49]. Together, these studies implied that the metabolic and fiber-type transcription program is tightly coupled.

As for humans, muscle fiber type transitions that have long been noticed can adapt to various types of exercise. Previous studies have shown that endurance exercise, which consumes more oxygen than resistance exercise, tends to increase the proportion of oxidative fibers [50,51,52]. The nuclear abundance of Pgc1α was increased after a 90 min cycling compared to the biopsy extracted before exercise in eight healthy young men (age 29 ± 3 years) active in endurance exercise training [53], supporting the conserved signaling pathway regulating mitochondrial biogenesis and exercise adaptation in mammals. The effect of resistance exercise itself on mitochondria remains inclusive. Most studies support that the mitochondria content is decreased upon resistance exercise due to increased muscle fiber cross-section area as shown by lower SDH staining intensity [54,55] and reduced numerous enzyme activities, including citrate synthase [56,57], hexokinase, myofibrillar ATPase, citrate synthase, phosphofructokinase, myokinase and creatine kinase [58], termed “resistance training-induced mitochondria dilution”. 

However, other studies reported that resistance exercise does not affect mitochondria content, although similar techniques are used to measure them [59,60,61]. The discrepancy might result from the different resistance exercise types (combined eccentric and concentric or concentric resistance training), especially muscle hypertrophy, which was not observed in the studies [59,61]. Due to gender differences, only women were recruited in the second study, which might have lower gains in muscle mass response to resistant exercise compared to men and could confound the interpretation of the result [60].

Despite the inconsistent findings on the mitochondria volume, the effect of resistance exercise on mitochondria function seems to reach a consensus that resistance exercise training can increase the mitochondria respiration activity in human skeletal muscle [62,63]. Eleven young, healthy, and physically active males (age 26 ± 5 years) were recruited to a 12-week resistance exercise training at an intensity of 60–80% of one-repetition maximum, the maximal weight an individual can lift for only one repetition [63]. There was a marked increase in lean body mass by 4% and the vastus lateralis muscle by 15%, as well as in muscle strength postexercise. Coupled mitochondrial respiration, particularly complex I, was increased in permeabilized muscle fibers, correlated with increased complex I abundance in the vastus lateralis muscle after training. Similarly, the long-term resistance-trained males (age 25.4 ± 6.1 years, n = 11) have higher state three mitochondrial respiration in permeabilized vastus lateralis fibers than nontrained but physically active males (age 25.4 ± 3.8 years, n = 11) [62]. Their maximal pulmonary O_2_ uptake (VO_2max_) and vastus lateralis muscle fractional O_2_ extraction are also greater during an acute incremental cycle ergometer. Thus, resistance exercise training led to a conspicuous increase in functional skeletal muscle mitochondrial respiration.

### 2.2. Mitochondria Oxidative Stress in Different Muscle Fiber Types

Although the primary function of mitochondria is energy production in the form of ATP, mitochondrial ROS generation is also essential for cellular homeostasis, which serves as a secondary messenger of various cell processes, such as cell renewal and differentiation. ROS functions as a key factor in activating various transcription programs [64,65] and glucose transport [66,67,68] in the muscle upon exercise stimuli. Treating antioxidants or inhibitors of ROS production attenuates exercise-mediated cellular adaptation [69]. As a side product of the mitochondrial oxidative phosphorylation process, mitochondria are consistently loaded with ROS. The overload of ROS is regarded as the primary cause of mitochondria dysfunction, contributing to mtDNA mutation, mitochondria membrane potential decrease, and mitochondria calcium accumulation [70]. The level of ROS is tightly controlled by generation and antioxidant defense systems in the cell. The general antioxidant defense systems comprise (1) antioxidant molecules that function through scavenging ROS, neutralizing the oxidative products, or inhibiting ROS generation, and (2) antioxidant enzymes, which decompose ROS. 

To manage cellular ROS at the physiological level, mitochondria have developed as a central hub of antioxidant defense systems in the cell. In the simplified formula, superoxide dismutase (SOD) converting radical anion (O_2_•−) to hydrogen peroxide (H_2_O_2_) and molecular oxygen (O_2_), hydrogen peroxide (H_2_O_2_) will then be converted to water by glutathione peroxidase, peroxiredoxin, and catalase, therefore reducing ROS emission from mitochondria. Although glycolytic muscle fibers have ~50% fewer mitochondria compared to oxidative muscle fibers in rats, the free radical leak (H_2_O_2_ production/O_2_ consumption) is two-to-three-fold higher in permeabilized fiber bundles from glycolytic muscles (white GAS) than from oxidative muscles (soleus or red GAS) [71]. Consistently, the scavenging capacity of exogenous hydrogen peroxide is higher in the soleus, followed by red GAS, and lowest in white GAS, suggesting that mitochondria have the ability to decompose ROS from other cellular sources. It has been shown that mitochondria isolated from rat brains [72,73,74] or skeletal muscles [75,76] are able to eliminate exogenous hydrogen peroxide in the presence of respiratory substrates, suggesting that ROS scavenging is coupled with activated respiratory processes. The greater ROS scavenging capacity and lower oxidative stress reported above might be due to more antioxidant enzyme expression in oxidative muscle, mostly resulting from higher content mitochondria. Since the normalization of glutathione peroxidase activity to the mitochondrial content, glutathione peroxidase activity is no different between glycolytic muscle and oxidative muscle fiber [71]. 

Increased ROS level has been shown to activate uncoupling proteins, which catalyze the transport of protons across the mitochondrial membrane and dissipate energy, thus minimizing ROS generation in a feedback loop. Uncoupling protein isotype 3 (Ucp3) is the major uncoupling protein selectively expressed in skeletal muscle [77,78]. Mitochondria lacking *Ucp3* produce more ROS isolated from the skeletal muscle of whole-body *Ucp3* knockout mice [79]. The markers of oxidative damage, including N^ε^-(carboxymethyl)lysine (derived from protein adducts of lipid peroxidation product) and aminoadipic semialdehyde (major carbonyl products of metal-catalyzed protein oxidation), were significantly increased in mouse skeletal muscle lacking *Ucp3* [80]. Using immunostaining to quantify Ucp3 expression level in the vastus lateralis muscle of young men, Ucp3 expression was shown to be highest in type IIx fiber and lowest in type I fiber [81,82]. The fiber-type specific Ucp3 expression might serve as an alternative way for glycolytic fiber to cooperate with ROS generation. 

Although Ucp3 could attenuate ROS generation, it, on the other hand, lowers respiratory efficiency. Skeletal muscle mitochondria lacking *Ucp3* reduced uncoupled respiration (state IV respiration), while coupled respiration driven by oxidative phosphorylation (state III respiration in the presence of ADP) does not change, thus augmenting the state III/IV ratio [79]. Studies have found that endurance exercise training known to promote mitochondrial biogenesis and oxidative fiber transition lowers Ucp3 expression in skeletal muscle, which is associated with increased VO_2max_ in young, healthy males [83,84,85,86]. Ucp3 expression is lower in all muscle fibers from the endurance-trained individuals (age 23 ± 5 years, n = 8) compared to the untrained but physically active counterparts (age 22 ± 3 years, n = 10), and the decrease of Ucp3 proteins was greater in the oxidative fibers than glycolytic fibers [85]. Since endurance exercise has been shown to recruit most slow-oxidative fibers (type I) after training, a subsequent study that includes high-intensity sprint training is known to recruit most fast-oxidative fibers (type IIa) for the comparison of fiber-type-specific reduction in Ucp3 [82]. Thirteen healthy and physically active young males (age 31 ± 6 years, n = 13) were recruited to a 6-weeks of either endurance (two interval sessions consisting of five to six intervals at an individual intensity corresponding to 70–80% of VO_2max_ and one constant intensity session for 40 min at an intensity of 60% of VO_2max_ per week) or sprint training (three sessions comprising 16–24 series of four to eight repetitions covering a distance of 40–80 m per week). Following training, endurance and sprint training had a comparable effect on decreasing Ucp3 transcription and protein levels in vastus lateralis muscles. While Ucp3 reduction was preferentially in type I after endurance training, the relative Ucp3 reduction was similar in all fiber types after sprint training, reflecting the fiber-type recruitment pattern by the respective training intensities. A lower Ucp3 content after training may be an adaptation that increases the efficiency of mitochondrial-coupled respiration and improves, in part, performance.

Whether reducing the ROS levels with antioxidant supplements could benefit muscle health has been a long-standing debate. In skeletal muscle, the combination of vitamin C (1000 mg/day) and vitamin E (400 IU/day) supplements abolished the improved insulin sensitivity induced by exercise training (a 4-week intensive exercise training) in healthy young men (age 25–35 years, n = 40) [87]. Molecularly, the exercise-induced expression of genes involved in mitochondrial biogenesis and ROS scavenging were also blocked by antioxidant supplementation. Consistently, two double-blinded, randomized controlled trials indicated that antioxidant supplementation blunts molecular adaptations to endurance exercise, as well as to strength training. Fifty-four healthy young adults (aged 20–30 years, 28 females and 26 males) were recruited to an 11-week endurance running training [88]. Prior to this study, 40 participants were engaged in regular regimes of endurance exercise, while the other 14 had less frequent weekly exercise. Despite the block of endurance training-induced expression of mitochondrial cytochrome c oxidase subunit 4 (COX4) protein in vastus lateralis muscle, the daily supplements of vitamin C (1000 mg) and vitamin E (235 mg) did not diminish training-induced systemically VO_2max_ and running performance (20 m shuttle run test). The second study on strength training involved 10 weeks of heavy-load traditional strength training at four weekly exercise sessions [89]. Thirty-two strength-trained volunteers (age 20–30 years, 11 female and 21 males) who were routinely engaged for one to four sessions of strength exercises weekly were recruited for this study. The muscle hypertrophy induced by strength exercise was not blocked; however, the gain of muscle strength was attenuated in the group with daily supplements of vitamin C (1000 mg) and vitamin E (235 mg). Collectively, these studies highlighted the significance of cellular ROS at a physiological level for muscle health and performance.

### 2.3. Mitochondria Calcium Uptake in Different Muscle Fiber Types

Calcium plays multifactual roles in regulating muscle functions. For example, calcium alone is sufficient to trigger muscle contraction. The sarcoplasmic reticulum is a specialized type of endoplasmic reticulum regulating calcium homeostasis and is a primary calcium storage site in skeletal muscle. Mitochondria is another site to manage cytosolic calcium concentration. It was first demonstrated that mitochondria isolated from rat kidneys could take up calcium in an activated respiration-dependent manner [90,91]. Mitochondria are later found to tether with the sarcoplasmic reticulum network via mitochondria-associated membranes, where calcium enters mitochondria from the sarcoplasmic reticulum [92,93]. Calcium crosses the mitochondrial outer membrane by calcium-permeable channels and voltage-dependent anion channels (VDACs), and transports into the mitochondrial matrix by a highly selective channel, mitochondria calcium uniporter (Mcu). During muscle contraction, calcium fluxes into mitochondria to activate enzymes involved in the TCA cycle, including FAD-glycerol phosphate dehydrogenase [94], pyruvate dehydrogenase, NAD-isocitrate dehydrogenase and oxoglutarate dehydrogenase [95] and electron transport chains responsible for oxidative phosphorylation, including complex III and V, thereby increasing ATP production. In order to keep intra-mitochondrial calcium at the physiological level, there are two types of cation exchangers exerting calcium efflux: the Na^+^/Ca^2+^/Li^+^ exchanger (NCLX), and the H^+^/Ca^2+^ exchanger, leucine zipper-EF-hand containing transmembrane protein (LETM1). Impaired mitochondrial calcium efflux causes calcium accumulation in mitochondria. Mitochondria calcium overload induces a collapse of the mitochondrial membrane potential, leaving the sustained opening of the mitochondrial permeability transition pore (mPTP), a non-selective channel across the inner and outer layers of the mitochondrial membrane. Consequently, pro-apoptotic factors, such as cytochrome c, are released from the mitochondria matrix through mPTP into the cytosol, triggering apoptosis. Given the fact that calcium homeostasis is fundamental for skeletal muscle function, skeletal muscle has developed its unique mitochondrial calcium uniporter complex containing an alternative splice isoform of mitochondrial calcium uptake 1 (Micu1), which can activate Mcu at lower calcium concentrations in the cytosol, thereby sustaining mitochondrial calcium uptake and ATP production [96]. The loss function mutation of *Micu1* was identified in a cohort of children suffering from muscle weakness and progressive involuntary movements with elevated serum creatine kinase levels, indicating skeletal muscle damage or degradation. The excessive mitochondrial calcium uptake was confirmed in the fibroblasts isolated from those young patients [97,98]. Impaired mitochondrial calcium homeostasis contributes to the pathophysiology of muscular dystrophy.

Recent studies using muscle-specific *Mcu* deletion mice showed impaired mitochondria calcium uptake, lowered capacity for treadmill running, and shifted metabolism toward fatty acid oxidation in skeletal muscle due to reducing the activity of pyruvate dehydrogenase, one of the enzymes activated by calcium [10,99]. Using MyoD-Cre, Cre recombinase, under the control of the MyoD locus that *Mcu* will be deleted from the progenitor cells of myofiber, did not affect muscle growth nor mitochondrial respiratory capacity [99]. In contrast, Gherardi et al., delete muscular *Mcu* using MLC1f-Cre, where Cre recombinase is only expressed in the differentiated muscle fiber, and its expression since embryogenesis reduces fiber size and respiratory rate in *Mcu*-deficient myofiber [10]. Even though these two papers have different results on respiration activities and muscle growth, which may be caused by different Cre recombinase used in the studies, the phenotypes of skeletal muscle observed in muscle-specific *Mcu* deletion mice are much more subtle than in the whole-body *Mcu* knockout mice [100]. The discrepancy between whole-body and muscle-specific knockout might be due to the outsourcing supplement of fresh mitochondria from the muscle regeneration process or other cell lineages requiring further investment. In contrast, mice with muscle-specific *Micu1* knockout have much more profound phenotypes in muscle dystrophy. Similar to patients who inherit *Micu1* mutations, muscular *Micu1* ablation by Ckmm–Cre, in which Cre recombinase under the control of creatine kinase promoter leads to *Micu1* deletion in fully differentiated skeletal muscle fibers and cardiomyocytes, causes muscle weakness and atrophy in mice. The increased serum creatine kinase level in *Micu1* muscle knockout mice is associated with impaired myofiber repair [101]. *Micu1* ablation myofiber has reduced calcium influx into mitochondria during contracting, which contributes to poor exercise performance. 

In terms of mitochondrial calcium management in the normal physiology of muscle fiber, mitochondrial calcium transporters and exchangers are highly expressed in oxidative fibers correlated with abundant mitochondria. A single-fiber proteomic study using vastus lateralis from young, healthy humans (aged 22–27 years, n = 4) reported ten-fold higher protein expression of Mcu in oxidative fibers than in glycolytic fibers [43]. Not only do oxidative fibers have more mitochondria, but the mitochondria isolated from the oxidative fibers also have a greater capacity for calcium uptake in rats and rabbits [102]. Moreover, nine weeks of neuromuscular electrical stimulation on the thigh elevated protein levels of Mcu in the elderly (aged 71.4 ± 7.1 years, n = 10). The increased Mcu protein level was accompanied by improved physical activities evaluated by maximal isometric torque and chair rise test (the time required to rise from a chair with arms folded across the chest) post-training [103]. Thus, mitochondrial calcium management is indispensable for muscle health. 

### 2.4. Mitochondrial Dynamics in Different Muscle Fiber Types

As a central signaling hub of the cell, mitochondria could quickly change their shape and subcellular distribution to adapt to cellular and environmental remodeling. For example, in the depletion of nutrition or other conditions triggering autophagy, mitochondria can be elongated via fusion to protect themselves from being degraded and maintain energy production. Starvation treatment is a common way to induce autophagic activity in vitro. It has been shown that various mammalian cells, including mouse myoblasts, treated with Earle s Balanced Salt Solution (EBSS) (contains 1% glucose) for amino acid or serum growth factors starvation for 1 h induce mitochondria elongation [104]. Interrupting the mitochondrial elongation process under nutrient depletion results in a loss of mitochondria. However, autophagy activation is not necessary for the mitochondria elongation upon nutrition stress. Knockout of autophagy-related 5 (*Atg5*), which is indispensable for autophagic vesicle formation, did not abrogate mitochondria elongation in cells under starvation [105]. 

Mitochondria in skeletal muscle form a dynamic network, named mitochondrial reticulum, to minimize metabolite distribution and maximize energy utilization efficiency [106]. The mitochondrial reticulum is constantly reshaped by fusion and fission events, allowing mitochondria to exchange their content, including mitochondrial DNA (mtDNA) [107]. The experiment, using fluorescent protein targeted to the mitochondria matrix in the established embryonic rat cardiomyocyte, showed the rapid redistribution of fluorescent protein from one mitochondrion to the other [108]. 

The re-shaping of the mitochondrial reticulum is largely mediated by the dynamin superfamily, membrane-remodeling GTPases. Mitofusin (Mfn) controls the fusion of mitochondria’s outer membrane, and optic atrophy-1 (Opa1) is required for the fusion of mitochondria’s inner membrane, in which the process results in two or more mitochondria merging and forming elongated mitochondria reticulum [109,110]. In cells, Opa1 exists in two forms: a membrane-anchored long form (L-Opa1) located in the inner membrane of mitochondria and a short form (S-Opa1) that lacks the transmembrane region derived from the cleavage of L-Opa1 by mitochondrial metalloendopeptidases: M-AAA protease 1 (OMA1) or YME1 Like 1 ATPase (YME1L) enriched in the intermembrane space [111,112]. While L-Opa1 is responsible for mitochondrial fusion, S-Opa1 is responsible for the maintenance of mitochondrial cristae structure. 

Under normal physiology conditions, mitochondrial fusion allows the cell to build an interconnected mitochondria network to promote oxidative metabolism. Mitochondrial fusion also facilitates the mixing of content, providing complementation for mitochondria with mutant mtDNA from mitochondria with normal DNA. It has been documented that cells could tolerate up to 60–90% mutant mtDNA without affecting the mitochondrial respiration rate or ATP synthesis [113]. For example, MERRF (myoclonic epilepsy and ragged-red fiber disease) patients carrying a mutation at the mitochondrial tRNA^lys^ gene only present the clinical syndromes when mutant mtDNA reaches 90% of total mitochondria in skeletal muscle [114]. Moreover, the increased size of mitochondria by fusion prevents mitochondrial elimination via mitophagy [115]. 

Mitochondrial fission, in contrast, is the process of one mitochondrion dividing into two mitochondria. The fission process is regulated by cytosolic dynamin-related protein 1 (Drp1). In mammals, Drp1 is recruited to the scission sites of the mitochondrial outer membrane by mitochondrial fission 1 (Fis1), mitochondrial fission factor (Mff), and mitochondria dynamic proteins of 49 and 51 kDa (Mid49/51). Subsequently, Drp1 polymerizes into spirals around mitochondria through GTP hydrolysis, allowing dynamin 2 (DNM2) to execute membrane scission [116]. During cell division, mitochondrial fission ensures the number of mitochondria is enough to segregate between daughter cells. In addition, when the dysfunctional components cannot be repaired, mitochondrial fission can segregate the damaged part and target it for degradation, thereby preserving the healthy mitochondrial network [116]. Mice with defects in mitochondria fission by acute knocking down *Drp1* or *Fis1* were reported to increase damaged mitochondria accumulation in adult skeletal muscle, which contributes to muscle degeneration and weakness [117,118,119]. *Drp1* deficient mitochondria are large in size and increase in calcium uptake, indicating that dynamic reshaping of the mitochondrial architecture ensures mitochondrial quality is under control [117].

Mitochondria morphology is highly related to the function of mitochondria. For instance, mitochondria in oxidative fibers are found to be more filamentous, while mitochondria in glycolytic fibers are more fragmented (Figure 1) [120,121]. Endurance exercise not only promotes the oxidative capacity of skeletal muscle but also modifies the expression of factors that manage mitochondrial dynamics. A study recruited 11 trained male cyclists in middle age (age 36 ± 4.9 years) to perform a 10 km cycling trial with an increase in altitude from 500 to 1250 m. Muscle biopsy was obtained prior to the trial, immediately after the trial, and post-exercise (2 h and 24 h). Transcription of mitochondrial outer membrane fusion regulators *Mfn1* and *Mfn2* are significantly induced 24 h after an acute exercise bout [122]. Another study recruited 17 older adults (age 66 ± 1 years, ten male and seven female) previously sedentary for 12 weeks of aerobic exercise intervention. Muscle biopsy was obtained prior to and after the intervention. The maximal VO_2_ is significantly increased following the intervention. The transcription of *Opa1* and *Drp1* is significantly up-regulated after exercise intervention, and *Mfn1/2* only exhibits a trend towards an increase. Despite the increased *Drp1* transcription, the total protein of Drp1 was not changed upon exercise training. In contrast, the phosphorylation of Drp1 at ser616 is reduced, which refers to a lower Drp1 activity in mediating mitochondrial fission [123].

Whole-body knockout of *Opa1*, *Mfn1/2*, or *Drp1* in rodents is embryonic lethal, strengthening the essential roles of fusion and fission processes during embryonic development [124,125,126]. Maintaining mitochondrial network dynamics is critical for metabolically active tissues such as skeletal muscle. Muscle-specific ablation of those genes that regulate mitochondrial dynamics has shown various degrees of severity in myopathy, and some even cause premature death in mice.

Muscle-specific KO mice for *Mfn1* and *Mfn2*, profusion factors mediating fusion of mitochondria outer membrane, are viable but die prematurely by 6–8 weeks of age [127]. Using MLC1f-Cre, where Cre recombinase is only expressed in the differentiated muscle fiber to excise *Mfn1* and *Mfn2* flox alleles, *Mfn1/2* null myofibers are small, and displays type IIb to IIa fiber transformation in TA muscles with the accumulation of damaged mitochondria. *Mfn1/2* null mitochondria lose their genomic DNA and increase point mutation. The ultrastructural analysis by electron microscopy reveals that *Mfn1/2* null mitochondria are fragmented into round spheres, and their cristae are dilated. Similar results have been observed in acute double deletion of *Mfn1/2* in adult skeletal muscle. Crossing *Mfn1* flox and *Mfn2* flox mice with transgenic mice expressing HSA-CRE-ER where Cre recombinase is driven by human skeletal actin (*HSA*) promoter and activated in the presence of tamoxifen, mice with tamoxifen-induced double deletion of *Mfn1/2* in adult skeletal muscle were generated. Tamoxifen was administered via intraperitoneal injection (2 mg/day for 5 consecutive days) in 3–5 months-old female mice. No lethality was reported in these mice, but poor exercise performance was observed starting 2 weeks after tamoxifen administration [128]. 

A more severe phenotype, neonatal lethality, was observed in mice with muscle-specific ablation of *Opa1*, a profusion factor mediating the fusion of mitochondria inner membrane. Mice with skeletal muscle-specific deletion of *Opa1* were generated by crossing the *Opa1* flox mice with transgenic mice expressing MLC1f-Cre [129]. These mice die within 9 days of postnatal life accompanied by hypoglycemia and severe growth retard with disorganized sarcomere arrangement in myofibrils. Similar to *Mfn1/2*-deficient mitochondria, *Opa1*-deficient mitochondria are also smaller with dilated cristae. The mitochondrial supercomplex was unstable, and respiration activity was reduced in isolated *Opa1*-deficient mitochondria compared to the control. In mice, postnatal skeletal muscle growth occurs rapidly in the first three weeks after birth. For example, the number of myofibers and the total number of myonuclei per myofiber in mouse EDL muscles are finalized at the postnatal day 7 and day 14, respectively [130,131]. To overcome the deleterious phenotype of muscle-specific *Opa1* deletion during postnatal development, which might compromise the analysis in elucidating the role of Opa1 in mature myofiber, Tezze et al., further study *Opa1* deletion in the adult skeletal muscle from 5-month-old mice. Briefly, the *Opa1* flox mice were bred with transgenic mice expressing HSA-CRE-ER to generate inducible muscle-specific *Opa1* knockout mice. Tamoxifen was added to food chow for 5 weeks. Based on the food intake measurement, roughly 1 mg of tamoxifen was taken per mouse per day. We are not sure how earlier *Opa1* is fully ablated in the skeletal muscle, yet the reduction of whole-body weight due to decreased muscle mass could be observed after 4 weeks of tamoxifen treatment initiated. *Opa1* deletion in adult skeletal muscle induces muscle loss and weakness with mitochondrial dysfunction described above. The increased mtROS production from *Opa1*-deficient mitochondria triggers ER stress, contributing to progressive muscle catabolism. Elevated systemic pro-inflammation and aging features, such as white hair and kyphosis, were present in those mice. Most of them died within three months after initiation of tamoxifen administration to induce *Opa1* deletion. Blocking oxidative stress by Trolox, a vitamin E analog with potent antioxidant action, or MiTo-TEMPO, a mitochondrial-targeted ROS scavenger, restores normal ER function and reduces muscle atrophic transcriptome but does not rescue mitochondrial dysfunction in the inducible skeletal muscle-specific *Opa1*-KO mice. Confirming that increased mtROS production from *Opa1*-deficient mitochondria triggers ER stress and contributes to progressive muscle catabolism. Increased ER stress leads to fibroblast growth factor 21 (FGF21) induction, contributing to muscle atrophy and weakness under stress. Moreover, increased secretion of Fgf21 from skeletal muscle has been shown to act on white adipose tissue, causing lipolysis. Strikingly, knockout of *Fgf21* simultaneously with *Opa1* in adult skeletal muscle partially prevented muscle wasting and liver steatosis, restored glycemia and inflammatory cytokines to the normal level, and, most importantly, rescued the lethal phenotype. Despite the fact that mitochondria dysfunction was still present, the genetic data confirms that increased secretion of Fgf21 from muscle is responsible for systemic inflammation and lethality in the inducible skeletal muscle-specific *Opa1*-KO mice [129]. In contrast, transgenically induced Opa1 expression to a moderate level (~1.5-fold increase compared to control) protected mice from denervation-induced muscle atrophy and mitochondrial dysfunction [132]. The protection effect of Opa1 induction is long-term and broad since a moderate increase in Opa1 expression protected from muscle atrophy and extended lifespan in two independent mitochondrial disease mouse models (the whole-body knockout for the complex I subunit, *Ndufs4*, and muscle-specific knockout of the complex IV assembly factor, *Cox15*) [133]. Together, these genetic studies indicated that mitochondrial fusion is necessary for mitochondria function maintenance and thus contributes to skeletal muscle health and performance. 

Defects of mitochondria fission, on the other side, result in similar phenotypes in mitochondrial dysfunction as the defects of mitochondria fusion despite their opposite function in regulating the mitochondria dynamic. Muscle-specific loss of *Drp1*, a factor mediating mitochondrial fission, causes muscle wasting and mitochondrial dysfunction [117]. *Drp1* deficient mitochondria are bigger in size and frequently display damaged features due to impaired degradation targeting damaged mitochondria. Interestingly, constitutive *Drp1* ablation (by MLC1f-Cre) specifically affected glycolytic fibers (type IIx/IIb) that the number and cross-sectional area of fibers are reduced, while induced *Drp1* ablation (by HAS-CRE-ER) in adult skeletal muscle from 5-month-old mice show the atrophy in all the different fiber types. Similar to muscle-specific loss of *Opa1*, a profusion factor for the mitochondrial fusion of inner membrane, in mice [129], to a lesser extent, constitutive muscle *Drp1* ablation in mice leads to premature death (within 30 days of postnatal life). Yet, there is no report of death in an inducible model. Even though *Opa1* or *Drp1* deletion in skeletal muscle shared phenotype similarity, including increased ER-stress and ubiquitin-proteasome system, the systemic inflammatory response was absent in *Drp1* muscle knockout mice despite the Fgf21 production and secretion from muscle being high in both models. Favaro et al., suggest that the higher level of Fgf21 secretion might impact a broad and severe phenotype systemically in muscle-specific *Opa1* deleted mice. The other phenotype discrete between *Drp1* and *Opa1* deficient mitochondria is calcium uptake. The protein expression of Mcu, the mitochondrial calcium uniporter, is elevated in *Drp1* deficient mitochondria, accompanied by increased mitochondrial calcium uptake during high-frequency electrical stimulation. In principle, the consequence of mitochondrial calcium overload would reduce cytosolic calcium for muscle contraction and trigger apoptosis, contributing to muscle weakness and myofiber loss, respectively. The local inhibition of *Mcu* by acute knockout via transfection with electroporation of shRNA against *Mcu* or prolonged deletion via intramuscular injection of adeno-associated virus (AAV) with RNAi against *Mcu*, could restore cytosolic calcium level or reduce the amount of center-nucleated fibers, a marker for cycles of degeneration and regeneration, in inducible *Drp1* deleted adult skeletal muscle. Certainly, Drp1 is essential for mitochondria health, yet uncontrolled Drp1 expression, which disturbs mitochondrial dynamics, also impairs muscle growth and homeostasis. Transgenic overexpression of *Drp1* in muscle progenitor cells activated from the embryonic stage leads to postnatal growth defects in skeletal muscle and poor exercise performance examined in treadmill tests [134]. *Drp1*-overexpressed mitochondria functioned normally, but their distribution in adult myofiber was altered. The inter-myofibrillar mitochondria were absent, and the oxidative capacity of mitochondria examined by SDH activity (mitochondrial complex II activity) was restricted to the fiber periphery in *Drp1* overexpressed myofiber. Regarding retard muscle growth, the glycolytic fibers were affected more than oxidative fibers by *Drp1* over-expression. Prolonged ER stress inhibited muscle anabolism, particularly the Atf4/eIF2a pathway, while muscle catabolism, including the ubiquitin-proteasome and autophagy-lysosome systems, did not differ. The similar phenotypes of muscle atrophy and poor performance from muscle-specific *Drp1* overexpression or deletion mice revealed that dynamic changes in mitochondrial morphology are critical for muscle health and performance.

Another mouse line targeting mitochondrial fission has been generated; muscle-specific knockout of *Fis1*, which is required for Drp1 recruitment to the mitochondrial outer membrane, only causes accumulation of larger and damaged mitochondria with dysfunction in oxidative fibers. [119]. Muscle-specific *Fis1* knockout mice were generated by crossing the *Fis1* flox mice with transgenic mice expressing Ckmm-Cre, whose expression was in fully differentiated skeletal muscles. Adult muscle-specific *Fis1* knockout mice have reduced endurance exercise capacity and increased inflammatory response after acute exhaustive exercise. The fiber-type-specific defects from different mitochondrial dynamics protein mutations suggest that the regulatory machinery of mitochondrial dynamics is tailored to fiber-type physiology in order to maintain a heterogenous population of mitochondria within skeletal muscle. As mentioned above, mitochondria in oxidative fibers are more filament, while glycolytic fibers are more fragmented. Dual deletion of *Mfn1/2*, profusion factors for outer membrane fusion, reduced the presence of elongated mitochondria in oxidative fibers, suggesting that abundant filament mitochondria observed in oxidative fibers might be due to the higher fusion activity [121]. Whereas glycolytic fibers were much more sensitive to the Drp1 expression, a fission factor for outer membrane fission, glycolytic fibers displayed profound defects from either over-expression or knockout of *Drp1* [66,117]. Drp1 recruitment to the mitochondrial outer membrane is the first step for mitochondrial fission, which could be mediated by at least four resident mitochondrial outer membrane proteins identified so far: Fis1, Mff, Mid49, and Mid51. The whole-body knockout of *Mff* showed the increased size of the mitochondrial domain in the glycolytic fibers [121] despite mice dying at 13 weeks of age from heart failure [135]. However, muscle-specific knockout of *Fis1* only affects the mitochondria function in oxidative fibers, not glycolytic fibers [119], suggesting that the regulatory machinery of mitochondria fission is different in oxidative fibers and glycolytic fibers. Thus, these genetic data suggested that mitochondria do not function equally in different skeletal muscle fiber types. The unique features of mitochondria closely correlate with the different contractile properties of oxidative fibers and glycolytic fibers.

### 2.5. Mitochondrial Degradation in Different Muscle Fiber Types

The contractile machinery and power production from skeletal muscle highly relied on functional mitochondria. Thus, the clearance of unhealthy mitochondria is vital for maintaining skeletal muscle health. Mitochondria turnover regulated by the mitochondria biogenesis and selective autophagic degradation of mitochondria, termed mitophagy, contributes to the mitochondria quality control. Mitophagy is initiated by recruiting autophagy receptors on the mitochondrial outer membrane that mark damaged mitochondria to be engulfed by autophagosomes for subsequent lysosomal degradation. Autophagy receptors have the classic tetrapeptide sequence W/F/YxxL/I/V binding to lipidated mATG8 family (Lc3/GABARAP), ubiquitin-like proteins conjugated to autophagosomal membranes. The lipidation of mATG8 occurs parallel to the cargo identification by autophagy receptors and requires the E1-like Atg7, the E2-like ATG3, and the E3-like ATG12–ATG5–ATG16L1 complex, which determines the site of lipidation. Autophagy receptors could either reside or be recruited at the outer membrane of mitochondria to initiate mitophagy. Two types of mitophagy have been identified to date. Ubiquitin-dependent mitophagy is primed by Pink1/Parkin activation. Under normal conditions, PTEN-induced serine/threonine kinase 1 (Pink1) is consistently imported into mitochondria via the translocase of the outer mitochondria membrane (TOM) complex and degraded by the mitochondrial protease, such as the intramembrane protease presenilin associated rhomboid like (PARL). However, under cellular stresses which depolarize the mitochondria membrane and cause mitochondria damage, the mitochondrial import of Pink1 is stopped. Pink1 is then stabilized and aggregated on the outer membrane of dysfunctional mitochondria, which Pink1 will recruit and phosphorylate E3 ubiquitin ligase Parkin (Parkin) and ubiquitin to promote ubiquitination of outer mitochondrial membrane proteins and further enroll autophagy receptors, such as optineurin (OPTN), calcium binding and coiled-coil domain 2 (CALCOCO2, also known as NDP52), Neighbor of BRCA1 (NBR1), Tax1 binding protein 1 (TAX1BP1), and sequestosome1 (SQSTM1, also known as p62) to the outer mitochondrial membrane for mitophagy initiation. In contrast, ubiquitin-independent mitophagy is activated by up-regulation of mitophagy-specific receptors, which are located at the outer membrane of mitochondria, such as BCL2 interacting protein 3 (Bnip3), BCL2 interacting protein 3 like (Bnip3L, also known as NIX) cardiolipin and ceramide to recruit Lc3 encapsulating mitochondria for lysosomal degradation. FUN14 domain containing 1 (FUNDC1) is a mitochondrial outer membrane protein and mitophagy-specific receptor involved in ubiquitin-independent mitophagy, whose activation is regulated by dephosphorylation by the mitochondria-localized phosphatase PGAM5 to promote mitophagy.

Mitophagy was shown to be induced by eccentric exercise to degrade damaged mitochondria. Lipidation of Lc3 and expression of Parkin, p62, and Bnip3 was increased in isolated mitochondria from muscles of 6-month-old female mice undergoing 3 days of a downhill treadmill exhaustion program [136]. Mitophagy reporters confirmed the activity of mitophagy induced by acute exercise with the combination of the MitoTimer, a marker for degenerated mitochondria, and Lamp1-YFP, a membrane marker of lysosomes [137]. MitoTimer is a DsRed1-E5 fluorophore targeted to the mitochondria matrix, which is sensitive to oxidation. It normally fluoresces green but shifts to the red spectrum upon oxidation, indicating degenerated mitochondria. The co-localization of red signal from MitoTimer and Lamp1-YFP marked degenerated mitochondria undergoing mitophagy were increased in muscle after an acute exhaustion exercise. Mitochondria quality control and mitochondrial dynamics are two interdependent processes. Mitochondrial fission is essential to isolate damaged mitochondria from the mitochondrial reticulum and thus facilitates mitophagy. Inhibition of fission or the promotion of fusion restricts mitophagy [104]. The phosphorylation of Drp1 at the ser616, which promotes Drp1 activities in mitochondrial fission, was transiently increased immediately following an acute exercise to facilitate mitophagy, followed by the phosphorylation of Drp1 at ser637, which inhibits its fission activity, 3–6 h post-exercise. Similarly, the mitochondrial recruitment of Parkin and its E3 ubiquitin ligase activities by examining Mfn2 ubiquitination was transiently induced in skeletal muscle following an acute exercise in young mice (3-month-old) [138]. Mfn2, a profusion factor for the outer membrane of mitochondria, was previously shown in cardiomyocytes to be the ubiquitination target for Parkin, functioning as a mitophagy receptor to recruit Parkin into the damaged mitochondria in a Pink1-dependent manner [139]. These data again reinforce that mitochondrial dynamics are tightly regulated in sustaining skeletal muscle health.

Exercise training (1 month of voluntary wheel running) in 3-month-old male mice has been shown to improve endurance capacity assessed by treadmill running test, that enhanced mitochondria quality was thought as a molecular outcome of exercise training [140]. Mitophagy-related proteins, including mATG8 member, Lc3, Atg7 required for mAtg8 lipidation, Beclin1, a class III PI3K that is essential for autophagosome formation, p62, an autophagy receptor, and Bnip3, a mitophagy receptor were examined in the muscle samples from mice 1-day post-exercise to avoid any acute effect of the exercise. The basal expression of Lc3, Beclin1, and Bnip3 alone with Lc3 lipidation are up-regulated, but p62 is down-regulated, and Atg7 has no change in the plantaris muscle with mixed fiber type upon exercise training. As for the soleus muscle (majority composed of oxidative fiber), Lc3, Atg7, Beclin1, and p62 alone with Lc3 lipidation are up-regulated, but Bnip3 has no change upon exercise training. Interestingly, although *Beclin1* heterozygous mice have comparable endurance capacity as control mice at the basal level, the exercise training failed to improve their endurance capacity further. The increase of Lc3 lipidation and induction of Bnip3 is also abolished in the plantaris muscle from *Beclin1* heterozygous mice following exercise training. This study implicated that increased basal autophagy/mitophagy is responsible for endurance capacity improvement through exercise training. A similar study conducted 1 h swimming exercise training 5 days/week for 8 weeks in 10-week-old male mice examined the flux of autophagy/mitophagy [141]. Mice were intraperitoneally injected with colchicine (0.4 mg per kg per day) to impair autophagosome–lysosome fusion on the last day of and one day after the final swimming session, and triceps muscle (composed of mixed fiber type) were harvested two days after the final swimming session. Autophagy flux examined by Lc3 lipidation and mitophagy flux shown by Bnip3 was increased upon exercise training. The basal level of Atg7 and Beclin1 is also up-regulated by exercise training. In this study, the mitochondrial profusion factor for the outer and inner membrane, Mfn2 and Opa1, respectively, and the mitochondrial fission factor for the outer membrane, Drp1, were examined and were all shown to increase in the basal level upon exercise training. Even though mitophagy flux was shown to be increased based on Bnip3 accumulation from the exercise group treated with colchicine, the active change of mitochondrial morphology will either promote or limit mitophagy, which interferes with the assessment of the absolute mitophagy activities. The net mitochondrial content is the sum activity of mitochondrial biogenesis versus degradation. Exercise training is known to increase mitochondrial biogenesis via activation of Pgc1α, but inhibition of autophagy via colchicine treatment blocks Pgc1α induction and thus decreases mitochondrial biogenesis in the exercise samples. Therefore, reducing mitochondrial content by colchicine treatment reflects a decrease in mitochondrial biogenesis instead of mitophagy flux. In order to further clarify the activity of mitophagy flux upon exercise training, Chen et al., examined the presence of Lc3 lipidation in the isolated mitochondria with or without colchicine treatment as a readout of mitophagy flux [142]. Mice at three months of age were trained by voluntary wheeling running for 6 weeks and were injected with colchicine (0.4 mg per kg per day) at the end of the training program for two days. On the last day of injection, mice were placed in a treadmill exhaustion test and harvested immediately after the test. Consistent with the previous finding, the acute exercise exhaustion test increased the flux of Lc3 lipidation and p62 association with mitochondria, indicative of enhanced mitophagy flux. However, the acute mitophagy flux induced by exhaustive exercise was attenuated in exercise-trained mice compared to the untrained mice. The attenuation of exhaustive exercise-induced mitophagy flux is mostly due to beneficial adaptation to cooperate with acute stress faster in the exercise-trained mice, which have increased functional mitochondrial number and mitochondrial recruitment of Parkin, leading to enhanced mitochondria quality control. 

The importance of mitophagy in muscle function was further validated in various mouse models with defects in general autophagy or specific mitophagy receptors. Muscle-specific deletion of Atg7, a noncanonical ubiquitin-activating enzyme E1 for mATG8 lipidation, causes muscle atrophy and weakness progressive with age [143]. Muscle-specific *Atg7*-knockout mice, generated by crossing the *Atg7* flox mice with transgenic mice expressing MLC1f-Cre, are viable but have a shorter lifespan [144]. The accumulation of giant mitochondria containing abnormal cristae in the Z-line, where the sarcomere ends, and the presence of swollen mitochondria were observed in *Atg7*-deleted EDL muscles associated with oxidative stress revealed by increased protein carbonylation. *Atg7* deficient mitochondria isolated from FDB myofibers were largely depolarized. A similar result on mitochondria defect was reported in another mouse line using Ckmm–Cre to excise *Atg7*, which decreased mitochondrial oxygen consumption [145], indicative of the importance of autophagy in mitochondrial quality control. Despite skeletal muscle and mitochondria dysfunction, no endurance running capacity defects were observed in 6-month-old muscle-specific *Atg7* knockout male mice [146]. Increased muscle Fgf21 secretion leads to white adipose atrophy and enhanced systemic glucose and insulin sensitivity in muscle-specific *Atg7* knockout mice. Deletion of *Fgf21* attenuates systemic metabolic protection against high-fat-induced metabolic dysfunction in those mice [146]. No muscle-type specific defects were reported in the above studies since autophagy is a general degradation process not specific for mitochondria but also long-lived proteins and other organelles in skeletal muscle. Moreover, even though Atg7-dependent autophagy is abolished, Atg7-independent alternative autophagy could compensate for sustaining a certain level of autophagy in *Atg7*-deficient muscle fibers for survival. Nonetheless, muscle degenerations and abnormal mitochondria accumulation were also observed in *Atg7* ablated adult skeletal muscle [143]. Two-month-old tamoxifen-inducible muscle-specific *Atg7* knockout female mice (*Atg7*flox/flox; HAS-CRE-ER) were administrated with tamoxifen via intraperitoneal injection (5 ug/day for one week). By two weeks of *Atg7* deletion, inhibition of Lc3 (mATG8 member) lipidation and accumulation of autophagy receptor, p62, was evidently observed in skeletal muscle. Acute deletion of *Atg7* leads to muscle atrophy, weakness, and accumulation of mitochondria in atrophic fibers without affecting endurance exercise performance. Although Atg7-dependent autophagy does not affect endurance running capacity, it is required for preserving mitochondria against exercise-induced mitochondrial damage [136]. Marco Sandri’s group using the same tamoxifen-inducible muscle-specific *Atg7* knockout mice further showed that acute inhibition of Atg7-mediated autophagy/mitophagy prior to exercise exacerbated mitochondrial depolarization and ROS generation following eccentric exercise in 6-month-old female mice [136]. The sex difference in muscle and mitochondria damage response to exercise is required for more follow-up. 

The whole-body knockout of *Parkin* leads to a decrease in state three mitochondria respiration in young mouse muscles (2–3 months old) [138]. The progressive mitochondrial dysfunction was further observed in the state four ROS production from old *Parkin* whole-body knockout mouse muscles (18 months). The exhaustive exercise-induced mitophagy flux was abolished in the young *Parkin* whole-body knockout mouse muscles despite no differences in total running distance between young control and *Parkin* whole-body knockout mice. Compared to *Atg7* knockout mice, the muscle phenotype observed from *Parkin* whole-body knockout mice is much more subtle could be due to the compensation of ubiquitin-independent mitophagy. Nonetheless, over-expression of Parkin has positive effects on increased mitochondrial activities and muscle hypertrophy in young male mice [147]. Young mice (3-month-old) were intramuscularly injected with AAV containing the sequence coding for *Parkin* driven by muscle creatine kinase promoter into the right TA and GAS muscles, whereases the contralateral leg (the left TA and GAS muscle) was injected with control AAV. Two AAV injections were applied to the mice at 3 and 5 months of age, and mice were harvested at 7 months. Increased complex II (SDH staining) and complex IV (COX staining) activities and fiber size and weight were observed in muscles injected with AAV–*Parkin* compared to AAV–Control. Similar results were obtained in older male mice (18-month-old mice injected with two AAVs at 18 and 20 months old and harvested at 22 months). In the older mice, the strength of TA muscle measured via in situ contractile stimulation was also improved in Parkin overexpressed TA compared to the contralateral TA. Collectively, these data suggest that Parkin protects against aging-induced oxidative stresses with respect to improving mitochondrial and, thus, muscle health.

As for the ubiquitin-independent mitophagy, mice with germline ablation of *Bnip3* [148], *Bnip3L* [149], or *Fundc1* [150] are normal and fertile. For normal development, mitophagy is critical for erythroid maturation, which eliminates mitochondria during differentiation, whereby inactivation of mitophagy is required for platelet survival, and both processes are highly dependent on Bnip3L activities [149,151,152,153]. Bnip3L null mice develop anemia and thrombocytosis due to mitochondrial retention in erythrocytes, impairing erythroid maturation and increasing platelet number by preventing Bnip3L-mediated autophagic degradation of mitochondrial protein Bcl-XL, which inhibits the activation of mitochondria-mediated apoptosis. Despite increased platelets, bleeding- or FeCl3-induced carotid artery thrombosis-induced platelet activation is impaired in *Bnip3L* null mice, reinforcing the importance of the functional mitochondria for full platelet activation, which is the energy-intensive process to sustain thrombus [151]. Regarding skeletal muscle, it has been shown that hypoxic-induced mitophagy (8% oxygen for 72 h) was attenuated in various tissues from *Fundc1* null mice, including skeletal muscle [150]. Skeletal muscle-specific deletion of *Fundc1* (*Fundc1* f/f; Ckmm–Cre or *Fundc1* f/f; HSA–Cre) mice diminished exercise performance with decreased maximal mitochondrial respiratory and ATP production upon exercise, while there was no change in muscle fiber types or mitochondria content reported in the study [154]. 

In addition to the conventional mitophagy described above, recent studies have reported the secretion of mitochondrial-encapsulated microvesicles (mitovesicles) to outsource mitophagy upon mitochondrial stress in various types of cells [155,156,157]. As for the skeletal muscle, it has been shown an increased secretion of mitochondrial markers in serum vesicle fraction from muscle-specific *Atg7* knockout mice (*Atg7f/f*; *Ckmm–Cre*, 4-month-old male mice) followed by a 3 days-consequent-eccentric-exercise to compensate for the mitochondria quality control [158]. Yet, those mitovesicles containing mitochondrial DNA and damage-associated molecular patterns could be a resource to activate innate immunity [156,159]. 

No direct evidence shows that mitophagy activity differs between different muscle fiber types, partially due to challenges in real-time monitoring and segregating mitochondrial biogenesis and degradation in vivo. With the intervention of mitophagy reporter mice such as mito-QC and mt-Keima, which targeted pH-sensitive fluorescence proteins to the mitochondria, the change of fluorescence activity could be used to follow the engulfing process of autolysosome, thus surveilling mitophagy flux in different muscle types is possible. Moreover, the newly developed mice that carried the MitoTimer reporter described earlier could differentiate newly synthesized mitochondria and oxidative mitochondria-targeted for degradation, which will also advance our knowledge in mitochondrial quality control in more detail.

## 3. Mitochondria in Human Myopathies

Mitochondria utilize various nutrients and generate diverse metabolites for cells to function and survive besides ATP. Hence, inherited mitochondrial diseases affect almost the entire body, yet those high energy-demand tissues such as skeletal muscle, cardiac muscle, and nerves are predominantly affected. The estimated prevalence of mitochondrial diseases ranged from 5 to 25 cases per 100,000 births [160,161,162]; however, there were no worldwide epidemiological data to verify these estimates. The clinical and genetic variability of the diseases makes it difficult to estimate prevalence accurately. Many patients with mitochondrial diseases may never have been diagnosed because of their nonspecific symptoms, leading to the masquerade of other diseases. Three forms of genetic alternation: mutations in the nuclear DNA encoded mitochondrial genes for mitochondrial morphology, dynamic function, biogenesis, or degradation; mutation or deletion of mitochondria DNA could all contribute to primary mitochondrial myopathies [163]. The updated sequence database from the mitochondrial diseases is available at MSeqDR (https://mseqdr.org/ (accessed on 10 June 2023)) [164]. There are also secondary mitochondrial diseases from mutations of nuclear DNA that indirectly affect mitochondrial operation. Primary mitochondrial myopathies are subtypes of inherited mitochondria diseases in which patients have prominent skeletal muscle problems, including fatigue, weakness, and exercise intolerance. This review will only focus on myopathy, even though other symptoms of primary or secondary mitochondrial diseases are present. The other manifestations have been discussed in other reviews, which readers can refer to [165,166].

### 3.1. Primary Mitochondrial Myopathies

The diagnosis of primary mitochondrial myopathies is confirmed by genetic tests sequencing variations or deletions in the genes known to cause the disease, in addition to histochemical and biochemical analysis in muscle biopsy. An elevated blood lactate level could also signify a deficiency in the electron transport chain. Detailed family history is also required to verify the pattern of inheritance. The major type of genetic mutation or deletion causing primary mitochondrial myopathies is summarized in Table 1. The decline of mitochondria respiration function due to genetic alternations contributes to skeletal muscle weakness, fatigue, and exercise intolerance in primary mitochondrial myopathies [167,168,169,170,171,172,173]. Mitochondrial myopathies can affect various muscles and cause muscle dysfunction all over the body, including facial and neck muscles, limb muscles, and even cardiac and respiration muscles, which may lead to death [173,174,175]. 

Chronic progressive external ophthalmoplegia (CPEO) describes a range of heritable myopathy that affects the extra-ocular muscle particularly. The common symptoms of CPEO are ophthalmoplegia, paralysis or weakness of the extra-ocular muscles, and ptosis, droopy eyelid. As the condition deteriorates, oropharyngeal and proximal muscle weakness emerge. The majority of patients tested possess at least one mitochondrial DNA deletion. The other frequent mutations on nuclear DNA encoded genes required for mitochondrial functions are genes involved in mtDNA replication: *POLG*, polymerase gamma, *C10ORF2*, mtDNA helicase, *RRM2B*, ribonucleotide reductase, and *TK2*, mitochondrial thymidine kinase 2; mitochondrial ATP transportation: *ANT1*, adenine nucleotide transporter 1; mitochondrial morphology: *Opa1*, profusion factor for mitochondrial inner membrane.

Thymidine kinase 2 deficiency is also responsible for Mitochondrial DNA Depletion Syndrome 2. TK2 is required for recycling nucleotides for mtDNA synthesis and repair. *TK2* mutation causes a shortage of nucleotide for mtDNA maintenance, leading to the depletion of mtDNA. The symptom of thymidine kinase 2 deficiency predominantly presents with muscle manifestations. The breathing difficulties usually cause respiratory failure and lead to early death in the infantile-onset form. Progressive muscle weakness from the proximal limb, ophthalmoplegia, and oropharyngeal are commonly reported, similar to CPEO.

Kearns–Sayre syndrome (KSS) is a neuromuscular disorder and is often characterized by pigmentary retinopathy with complications associated with CPEO. Progressive cardiac dysfunction and congestive cardiac failure are also features of KSS. The onset of KSS typically starts before age 20. KSS is predominantly caused by single large-scale deletions of mtDNA.

Mitochondrial encephalomyopathy, lactic acidosis, and stroke-like episodes (MELAS) are generally linked to complications in multiple body systems. Typical symptoms related to skeletal muscles are muscle weakness and muscle denervation. Excess mitochondria are accumulated in subsarcolemmal zones of skeletal muscle fibers. MELAS is caused by a point mutation in one or multiple mitochondrial encoded genes for mitochondrial tRNA: *MT-TL1*, *MT-TH*, and *MT-TV* and OXPHOS complex: *MT-ND1*, *MT-ND5*. The majority of cases (80%) have the m. 3243A>G mutation in the *MT-TL1*.

Myoclonic epilepsy with ragged red fibers (MERRF) is another disorder affecting many body parts. Particularly, MERRF is known to cause muscle cramps, weakness, and progressive stiffness. When staining muscle biopsy from MERRF patients with Gomori Trichrome, the ragged-red fibers are visible under the microscope. The accumulation of mitochondria results in an irregular outline of the muscle fiber, showing red ridges and a skewed sarcoma, which causes the “ragged” appearance. Mutations in mitochondrial tRNA for mitochondrial translation are the causes of MERRF. More than 80% of MERRF cases have mutations in *MT-TK*; the rest include *MT-TL1*, *MT-TH*, and *MT-TS1.*

### 3.2. Other Myopathies with Mitochondria Symptoms

The non-mitochondrial myopathies are elicited by mutations in other genes unrelated to mitochondria directly; some also have a diverse spectrum of mitochondrial symptoms. Myofibrillar myopathies (MFMs) are a group of rare genetic neuromuscular disorders that causes progressive muscle weakness involving skeletal, cardiac, and smooth muscles with various degree of severity. Myopathic conditions such as focal myofibrillar destruction and accumulated aggregates of myofibrillar elements define MFMs. Depending on the subtype and associated gene mutation, myofibrillar myopathies may manifest in childhood or late adulthood. The prime genetic mutation responsible for MFMs is outlined in Table 2. Most mutations are involved in intermyofibrillar scaffolds (located in Z-disc) for filamentous intermyofibrillar cytoskeleton arrangement, including intermediate filament proteins such as DESMIN, the actin-cross-linking proteins such as *MYOT*, *LDB3*, and *FLNC*, the molecular chaperon such as *CRYAB*, *BAG3*, and *HSPB8*, and the multifunctional cytoskeletal linker such as *PLEC*. Each sarcomere is connected by intermyofibrillar scaffolds (located in Z-disc) that anchor thin filaments and transmit force along the myofibril. During the contraction, the z-disc ensures myofibrils change length in unison and prevents damage to membrane systems that span between myofibrils. Intermyofibrillar scaffolds are also critical for the arrangement of the intermyofibrillar mitochondria located at the I-band, which align on both sides of the z-disc. MFMs are often underpinned by mitochondrial dysfunction at the cellular level. Myofibrillar destruction disrupts the distribution of mitochondria. Abnormal mitochondrial morphology, such as vacuolization and swollen, and mitochondrial mislocalization, such as the depletion of mitochondria in the intermyofibrillar space and aggregation in the subsarcolemmal space, have been reported in the muscles of MFM patients [181,182]. The study from mice lacking *Desmin* recapitulates many features in myopathy from MEM patients [183] and confirms the subsarcolemmal aggregation of mitochondria in young (1-month-old) and progressive in adult (8-month to 1-year-old) mouse muscle [184]. Noted that the isolated mitochondria from *Desmin* null mouse heart did not show respiratory defects in the present with pyruvate or glutamate plus malate or succinate alone. Yet, the single fiber isolated from the soleus or ventricular muscle of the heart displays a significant reduction of ADP-stimulated respiration, and the decrease of respiration is not due to mitochondrial content as citrate synthase activities were used as a quantification of mitochondrial content showed no difference. The respiratory defect in *Desmin*-null mice was not observed in fiber isolated from the GAS muscle, which has relatively lower mitochondria than the soleus muscle. Thus, these data implied proper mitochondrial positioning along muscle fibers is vital for effective reparatory activity.

Another common myopathy is central core disease (CCD), a rare genetic neuromuscular disorder classified as a congenital myopathy, which is present at birth (congenital) and causes muscle weakness. The typical histopathological feature of CCD is the appearance of “core” lesions, predominately in type I fibers. These cores are characterized by the depletion of mitochondria, which is validated by a lack of oxidative enzyme activities. Most cases of CCD carry mutations on Ryanodine receptor 1 (*Ryr1*), which encodes a calcium release channel in the sarcoplasmic reticulum [186]. Pathogenic *Ryr1* mutations result in calcium leaks from the sarcoplasmic reticulum, leading to mitochondrial calcium overload. Using transgenic mice to model Ryr1-related myopathy, Boncompagni and colleagues showed that the heterozygous Y522S knock-in (Y522S in human Ryr1 corresponding to Y524S in mouse Ryr1) mice showed the appearance of mitochondrial damage such as swollen, misshapen, and/or disrupted by EM analysis before the core formation at 2 months of age [187]. 

Other myopathies, including dermatomyositis, Pompe disease, metabolic myopathy, and dysferlinopathy (Table 3), are also found to show abnormal mitochondria morphology or/and mitochondria respiration activity defects [188,189,190,191]. In addition, increased ROS generation, mitochondria calcium overload, and mitophagy impairment have been reported in Pompe disease [189,192]. As described above, type I fiber predominance of mitochondria dysfunction is found in non-mitochondrial myopathies. Type I fiber-specific atrophy is also reported in several non-mitochondrial myopathies, implying that the high mitochondria content in type I fiber may cause it to be more susceptible to mitochondrial dysfunction [193,194].

## 4. Mitochondria in Skeletal Muscle Aging 

### 4.1. Effects of Aging on Mitochondria Content and Function

Mitochondrial content and respiration activity have been shown to decrease together with increased oxidative stress in skeletal muscle from both rodents and humans during aging [195,196,197,198,199,200]. Mitochondrial respiration measured in permeabilized muscle fiber shows decreased mitochondrial respiration with age from the vastus lateralis muscle of 38 participants recruited in the Baltimore Longitudinal Study of Aging (age 24–91 years) [199]. The age-related decrease in mitochondrial respiratory capacity was linked to declines in mobility (gait speed and time to complete the 400 m walking test), muscle strength (grip strength and knee extension strength), cardiorespiratory fitness (VO_2max_), and mitochondrial oxidative capacity (^31^P magnetic resonance spectroscopy estimates the phosphocreatine recovery rate after a rapid and intense ballistic knee extension exercise) independent of sex and body composition. Age-related decline in mitochondrial respiration activity reveals a pattern specific to muscle fiber type: the respiratory activity of mitochondria isolated from glycolytic muscle (TA) but not from oxidative muscle (soleus) decreased significantly in old mice (28–29 months of age) compared to their youth counterpart (3 months of age) [201]. Despite the similar aging-associated changes in proteins regulating mitochondrial homeostasis in oxidative and glycolytic muscle, oxidative muscles are protected from age-related mitochondrial dysfunction. 

Mitochondrial dysfunction, defined as lower respiration activity and greater oxidative stress, was speculated to drive cellular damage underlying muscle age [202,203]. An increase in mitochondrial oxidative stress resulting from ROS overproduction might trigger cell death and thus could be prevented by mitochondrial antioxidants. Despite this, there is still no consensus on whether ROS production is elevated in aging human muscle. Two-fold increased glutamate, malate, and succinate-induced H_2_O_2_ emission was reported in isolated mitochondria from old (age 67.3 ± 1.5 years, n = 6) than young (age 23.5 ± 2.0 years, n = 6) healthy sedentary men lived in Denmark [204]. Despite the lower systemically VO_2max_ in elderly subjects, no direct measurement of mitochondrial ATP production was reported from the above study. Opposite to the earlier study, a later study using a bigger cohort reported that both mitochondrial ATP and ROS production were significantly lower in old (age ≥ 65 years, n = 35) than in young (age 18–30 years, n = 22) healthy sedentary people in Texas, USA [205]. The decline of mitochondrial ATP and ROS production was consistent in the presence of different respiratory substrates: succinate, glutamate/malate, or pyruvate/malate. Yet, two other studies that recruited healthy men living in Denmark showed no age-associated difference in mitochondrial ATP and ROS production in the presence of respiratory substrates, succinate, or pyruvate/malate, despite the lower systemically VO_2max_. Notice that in the first study, the physical activities were comparable between young (age 20–30 years, n = 24) and old (age 60–70 years, n = 29) men that, on average, 3 days for hard exercise and 3–4 days for moderate exercise per week, which might hinder the age-related changes in this considerably activate healthy aging group [206]. Similarly, the daily physical activity level is also comparable between young (age 23.4 ± 0.5 years, n = 17) and old (age 68.1 ± 1.1 years, n = 15) men in the second study [207]. The lack of consensus on age-related changes in mitochondrial ROS production might have originated from the different physical activities of subjects between studies. 

Although there is no consensus on the effects of age on mitochondrial ROS production, increased oxidative damage shown by higher lipid peroxidation and protein carbonylation in aged skeletal muscle has been observed [208,209,210,211]. A previous study compared the activity of ROS scavenging by examining the expression of antioxidant enzymes and oxidative damage markers in rectus abdominis (muscle for posture) and vastus lateralis (muscle for movement) muscle from the elderly group (66–90 years old, n = 12~16) and young individual (18–48 years old, n = 5~8) [208]. A significant increase of lipid peroxidation was observed in aged vastus lateralis muscle, whereas no difference was found in aged rectus abdominis muscle compared to their respective younger counterparts. Among antioxidant enzymes, levels of Mn-dependent superoxide dismutase (SOD2) in rectus abdominis muscles of an older age group were higher than those of their younger counterparts and age-matched vastus lateralis muscle. Whether the selective protection of oxidative stress in aged rectus abdominis muscle due to muscle type transition with age is unclear since age-related reductions in type II fiber determined by histochemical analysis of myosin-ATPase were found in both muscle types. 

Meta-analysis of randomized controlled trials has shown that multivitamins used for antioxidant supplements had no effect on all-cause mortality [212] or prevention of aging-associated diseases [213,214,215,216]. In addition, as described in Section 2.2, even though excess ROS harms cells, physiological ROS levels are necessary for muscle health. Antioxidant supplements cancel the beneficial cellular adaptation to exercise training and, in some cases, may even impair muscle performance in young, healthy adults. Exercise is widely accepted as the most effective remedy for sarcopenia. In addition to augmenting mitochondrial biogenesis, exercise stimulates ROS generation as a byproduct resulting from the high respiration activity required for muscle force production [217]. Research has demonstrated that resistance exercise, particularly strength training, is effective in counteracting muscle weakness and physical frailty in very old people. The Boston FICSIT (Frailty and Injuries: Cooperative Studies of Intervention Techniques) study has shown that a 10-week supervised lower-extremity resistance training increases muscle strength by 113 ± 8 percent and muscle mass by 2.7 ± 1.8 percent, as well as improvement in various physical activities assessments in 100 frail nursing home residents (age 87.1 ± 0.6 years, 63 females and 37 males) [218]. Given the perception that older adults might be predisposed to oxidative damage and oxidative stress-related pathologies, numerous clinical trials have assessed whether combining antioxidant supplements and resistance exercise can magnify the efficacy of resistant exercise training on muscle health. A 6-month supervised resistance training (one-hour session for abdominal exercise and the upper and lower body strength training with three sets of eight repetitions at 80% of one-repetition maximum, the maximal weight an individual can lift for only one repetition) significantly increases muscle strength and fat-free appendicular mass in the healthy sedentary old adults (age 53–73 years, 30 females and 27 males) regardless of antioxidant supplements [219]. Moreover, there is no difference in lipid peroxidation markers or pro-oxidant status in serum from antioxidant supplements groups (1000 mg vitamin C and 600 mg vitamin E per day) with other groups. Another trial recruited an older male (age 61–80 years, n = 35) in a 12-week traditional strength training [220] as described in the previous study in youth [89] with modifications coordinating to older adults: weekly sessions reduced from four to three, the number of sets per exercise was progressively increased during the first 10 weeks and then reduced to one set on the last 2 weeks. There are no differences between placebo and antioxidant supplements groups (1000 mg vitamin C and 235 mg vitamin E per day) in muscle strength and muscle mass improvement with resistant exercise training. Yet, the placebo group achieved a greater increase of lean mass and larger growth of rectus femoris muscles postexercise compared to the antioxidant group. Since the gain of muscle mass has a major clinical significance, it is important to take precautions before the recommendation of antioxidant supplements for healthy elderly who are undergoing training in strength. 

Muscle weakness is one of the most noticeable aspects of sarcopenia. In addition to the slower turnover rate of myofibrillar proteins, particularly contractile proteins [221,222], altered calcium handling associated with increased oxidative stress has been reported in muscle aging. A significant reduction in excitation-contraction coupling was recorded in single muscle fiber isolated from vastus lateralis of old adults (age 65–75 years, n = 11) than young adults (age 25–35 years, n = 9) resulting from the decreased calcium release from sarcoplasmic reticulum for triggering muscle contraction [223]. A similar result has been observed in flexor digitorum brevis muscles isolated from old (age 20–22 months) versus young (age 3–6 months) mice [224]. The decreased calcium release from the sarcoplasmic reticulum is a result of lower calcium storage in the sarcoplasmic reticulum in vastus lateralis muscle fibers of old adults (age 70 ± 4 years, n = 20) than young adults (age 22 ± 3 years, n = 16) [225]. Reciprocally, the decrease in calcium influx into the myoplasm would ultimately hamper contraction-induced calcium uptake in the mitochondria. The reduced calcium uptake has been reported in an in vitro study on the myoblasts either harvested from GAS muscles of the young (6 months of age) or old (26 months of age) male mice [226]. Over-expression of mitochondrial calcium uptake family member3 (MICU3), which is reduced in aged muscle via an intramuscular injection of AAV9 encoding *Micu3* in old mice, restores the level of mitochondrial calcium uptake to their young counterpart. More importantly, the age-associated decline in ATP production and increases in ROS generation and apoptotic nuclei are also ameliorated in Micu3 over-expressed GAS muscle. Another potential cause affecting mitochondrial calcium uptake is reduced sarcoplasmic reticulum and mitochondria tethering, which has been reported in aged mouse skeletal muscles [227]. Disconnected sarcoplasmic reticulum and mitochondria have also been observed in healthy sedentary seniors (age 65–74 years, n = 9), whereas lifelong physical exercise (age 65–79 years, n = 15) improved the pairing of sarcoplasmic reticulum and mitochondria [228]. Maintaining the physical tethering between the sarcoplasmic reticulum and mitochondria minimizes the distance and duration of mitochondrial calcium import, which is crucial for the efficient production of ATP and the preservation of contractile properties in aged muscle.

### 4.2. Effects of Aging on Mitochondria Quality Control

In accordance with the age-related defects in mitochondria function, aging is also linked to abnormal mitochondrial dynamics and quality control in animal and human models. Mitochondrial dynamics is the continuous process of mitochondrial fission and fusion, building an interconnected network to maintain their integrity, morphology, and functions. The absence of reliable biomarkers to measure mitochondrial fusion and fission has hindered agreement on how aging impacts mitochondrial dynamics in skeletal muscle. The direct quantification of mitochondrial dynamics is by the traditional morphology analysis using electrical microscopes. Due to the sample preparation and the high mitochondrial heterogeneity in skeletal muscle described in Section 2.4, EM analysis limits the screening capacity. Alternatively, the expression of factors involved in mitochondrial dynamics was used as indirect readouts. However, the expression pattern of those factors during muscle aging was inconsistent between studies. Several reasons could account for inconsistent results in the expression of mitochondrial dynamic markers. First, fiber-type specific regulation in mitochondrial dynamics could affect the assessment of aging. Since the mitochondria in oxidative fiber are longer than in glycolytic fibers, the increase of mitochondrial pro-fusion factor, Mfn2, could be just a reflection of the increase in oxidative fiber proportion due to the reduction in glycolytic fibers in aged muscle. Second, multiple posttranslational modifications on mitochondrial dynamic regulators have been reported to either promote or impair their activities without affecting their stability. For example, the phosphorylation of Drp1 at ser616 by CDK1/CYCLINB or mTORC1 has been shown to promote mitochondrial fission, while the phosphorylation of ser637 by protein kinase A or Ca2^+^/calmodulin-dependent protein kinase Iα inhibits its GTPase activities, thus preventing mitochondrial division [229,230]. Lastly, either fission or fusion factors must be recruited to mitochondria to exert their function. Thus, their transcription or protein levels from the total muscle lysate could not be accurately indicative of their activities. 

Effects of aging on the protein expression level of mitochondria dynamic-related proteins (Fusion: Opa1, Mfn1, Mfn2; Fission: Drp1, Fis1) shows no significant difference between young and older adults [231,232,233], despite two studies that respectively reported significant decreases of Opa1 [234] and Mfn2 [235] in elderly. On the other hand, rodent studies show conflicting results due to differences in animal models (rats or mice), sex, muscle type, and age stage between studies (Table 4). Especially for the definition of aged animals, studies conducted in rats varied from 22 to 35 months [236,237,238,239,240,241,242], while studies performed in mice varied from 8 to 29 months [129,201,243,244,245,246]. In addition, the types of skeletal muscle included in these studies either contain glycolytic fiber predominant muscle (EDL, TA, triceps muscle, GAS, or quadriceps muscle), or oxidative fiber predominant muscle (soleus muscle), the difference in the muscle fiber type composition might underly the cause of conflict results. 

Published studies focus more on the glycolytic muscles (EDL, TA, White GAS) than the oxidative muscle (soleus). In glycolytic muscles, the increased protein level of Fis1 seems to reach a consensus among the studies list above [236,238,239,245], which might indirectly indicate that the mitochondria fission increases with aging in glycolytic fibers of rodents. Age-related protein expression level alteration of the other fission-related protein, Drp1, is inconclusive. It seems that the age-related changes of Drp1 are non-significant in mice, whereas it is up-regulated in rats. It seems that the age-related changes of Drp1 are non-significant in mice but up-regulate in rats. Since Drp1 is recruited to mitochondria by Fis1, the cytosolic level of Drp1 is not necessary to reflect the mitochondria fission activity. Thus, the study conducted in isolated mitochondria will be more convincing in addressing Drp1 activity.

As for mitochondria fusion, Opa1 protein expression levels show inconclusive results, as two papers show increased Opa1 protein levels or functional Opa1 isoform ratio during aging [201,236], while three other studies show reduced or no significant difference [129,238,245]. Crupi, Tezze, and Yeo’s studies are conducted in mice, but the aged group in Tazze’s (18 months old) and Yeo’s (24 months old) studies are younger than the aged group in Crupi’s study (28~29 months old), this might explain the different results obtained from those studies. O’Leary and Iqbal’s studies use the same age and sex rats; the difference in the result might be due to Iqbal’s study measured Opa1 expression in isolated subsarcolemmal mitochondrial fraction while O’Leary’s study measured in whole muscle lysate. Another fusion-related protein, Mfn2, also shows inconclusive results. All the studies conducted in mice show increased Mfn2 protein levels in aged glycolytic and oxidative muscle examined [201,245]. The different results from the three studies conducted in rats can also be explained by the different age and protein fractions examined in the studies, as discussed above [236,238,239].

It is worth pointing out that, among all those studies listed above, only three studies measure the changes of protein expression in mitochondrial fraction [201,237,238], while other studies are conducted in whole muscle lysate. Since those mitochondria dynamic-related proteins function in mitochondria, the results obtained from skeletal muscle mitochondrial fraction might be more direct evidence than those from whole muscle lysate. Two out of three studies using mitochondrial fraction from glycolytic fiber predominant muscle (GAS or TA muscle) obtained consistent results that fission-related proteins (Fis1 and Drp1) expression level is higher in aged rats than in the young rats [237,238]. The other study conducted in the TA and soleus muscles found that mitochondria fusion marker Mfn2 is significantly higher in aged muscles. No fission markers were looked at in the study; thus, a comparison between mice and rats could not be made [201]. Nonetheless, a recent study confirmed the age-related morphological changes in mitochondria from rat diaphragm muscle, consistent with mitochondrial fraction studies performed in rats that up-regulation fission markers in glycolytic fibers during aging. By performing and colocalizing MitoTracker staining used to label mitochondrial shape and MyHC staining used to differentiate muscle types in sequential sections from the rat diaphragm muscle, Brown et al., found that mitochondria are more fragmented in the aged glycolytic fibers [247]. However, another study conducted in mice, which used a better-established method (Transmission electron microscopy) to determine the mitochondria morphology, found that intermyofibrillar mitochondria of the GAS muscle (containing glycolytic fiber predominant muscle) are longer and more branched in aged mice in comparison with young mice. This study also reported an increase in the fusion protein (Mfn2) to fission protein (Drp1) ratio in whole GAS muscle lysate during aging, although the absolute level of Mfn2 and Drp1 show no significant difference [244]. The diverse results of these two morphology studies might be owing to the disparate species examined: rat versus mouse. Since the mitochondrial fraction study mentioned above in mice also observed higher expression of Mfn2 in aged muscles, it could be that the lengthy mitochondria are unique for muscle aging in the mice. Second, it also could be that age-related loss in glycolytic fibers causes oxidative fibers to be over-represented in aged GAS muscle. Even though GAS muscle is predominantly composed of glycolytic fiber, it is also bigger in size and does compose oxidative fiber. Without muscle type identification, making a conclusive correlation between mitochondrial dynamics and muscle type with aging is challenging. 

Mitochondria dynamic and mitophagy are closely related [248]. Muscle disuse induced by denervation mimics the mitochondria dynamic phenotypes observed in aging, such as the reduced ratio of fusion: fission proteins [238]. Denervation also increases the autophagic degradation activity in skeletal muscle; specifically, the protein level of the mitophagy marker, Parkin, is induced in both young and aged mouse muscles. Aged rats and mice have higher basal levels of Parkin and Pink1 [236,245], as well as Bnip3 [246,249], but their induction responses to denervation are lesser than young animals [236]. The reduced response to denervation-induced mitophagy in aged mice might cause damaged mitochondria accumulation and thus impair skeletal muscle function. The effects of aging on mitochondria dynamics and mitophagy are summarized in Table 4.

Taken together, mitochondria dynamic and mitochondria quality control play essential roles in regulating skeletal function and health. Mitochondria dynamic and mitophagy process seem to show age-related alteration, but more data from the oxidative fiber and mitochondrial fraction to study their activities are needed.

## 5. Conclusions

In this review, we critically summarize the current findings on mitochondrial functions related to skeletal muscle health, with a focus on in vivo studies from rodents and humans. We highlight the analysis of muscle characteristics reacting to changes in mitochondrial activities. Owing to higher mitochondrial numbers and activated mitochondrial dynamics, oxidative muscles are much more tolerant to age-related mitochondrial dysfunction and thus retain their integrity with age. Intervention targeting mitochondrial quality control to improve mitochondria quality or mitochondrial transplantation therapy to increase the number of functional mitochondria could be possible new therapeutic solutions to mitigate sarcopenia. 

## Figures and Tables

**Figure 1 cells-12-02183-f001:**
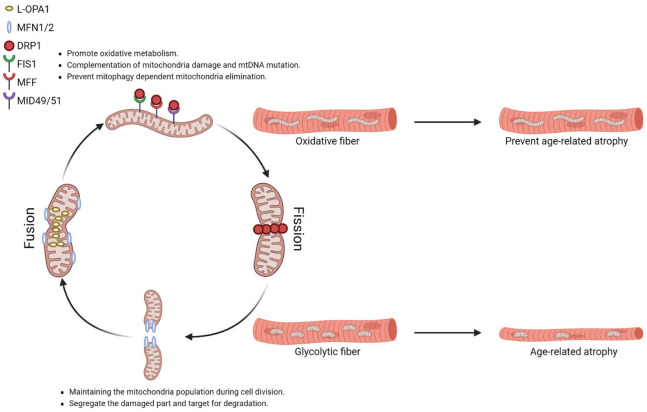
Molecular mechanism of mitochondrial dynamic and its relevant to age-related muscle atrophy. Mitochondria in skeletal muscle form a dynamic network and can be reshaped by fusion and fission process. Relationship between mitochondria dynamic and age-related muscle fiber atrophy is highlighted in this figure. Mitochondria in oxidative fiber is more filamentous which prevent the oxidative fiber from age-related loss of mitochondria and thus preserve the oxidative fiber with age. On the other hand, mitochondria in glycolytic fiber is more fragmented which cause glycolytic fiber to be affected more during aging.

**Table 1 cells-12-02183-t001:** The major types of genetic mutation or deletion cause primary mitochondrial myopathies.

Myopathy	Gene of Mutation	nDNA/mtDNA	Muscle and Mitochondria Phenotypes	Reference
CPEO	POLG1, POLG2, ATN1, C10ORF2, Opa1, TK2, and multiple mtDNA or mtRNA	nDNA/mtDNA	mitochondria with swollen cristae or paracrystalline inclusions. Reduce mitochondria respiration capacity. Increase ROS generation. Mitochondria fusion defect.	[167,174,176]
TK-2 DEFICIENCY	TK2	nDNA	Selective loss of type II fiber. Increase the proportion of SDH staining positive fiber and Cox staining negative fiber.	[177]
KEARNS-SAYRE SYNDROME	Variable single mtDNA deletion	mtDNA	Reduce mitochondria respiration activity.	[169]
MELAS	MT-TL1, MT-TH, MT-TV, MT-ND5, and MT-ND5	mtDNA	Enlarged mitochondria or slightly swollen small mitochondria. Reduce mitochondria respiration activity and increase ROS generation.	[168,178]
MERRF	MT-TK, MT-TL1, MT-TH, and MT-TS1	mtDNA	Reduce mitochondria respiration activity, increase ROS generation, ROS clearance defect.	[170,178,179,180]
COQ10 DEFICIENCY	COQ2, COQ4, COQ6, COQ7, COQ8A, COQ8B, COQ9, PDSS1, and PDSS2	nDNA	Reduce mitochondria respiration activity.	[171]
LEIGH SYNDROME	ND2, and SURF1	nDNA	Reduce mitochondria respiration activity, reduce mitochondria complex I activity.	[172]

**Table 2 cells-12-02183-t002:** The prime genetic mutation responsible for MFMs [185].

Myopathy	Gene of Mutation	nDNA/mtDNA	Muscle and Mitochondria Phenotypes
ab-crystallinopathy	CRYAB	nDNA	Increase the proportion of rubbed-out fibers (low complex II activity), Cox staining negative fibers (low complex IV activity), and paracrystalline inclusions.
BAG3 myopathy	BAG3	nDNA	Increase the proportion of rubbed-out fibers (low complex II activity)
Desminopathy	DES	nDNA	Altered mitochondria distribution. Cox negative fibers (low complex IV activity). Enlarged, vacuolated mitochondria. Mitochondria with abnormal cristae. Mitochondria with paracrystalline inclusion.
DNAJB6	DNAJB6	nDNA	Increase proportion of rubbed-out fibers (low complex II activity), Cox staining negative fibers (low complex IV activity)
FHL1	FHL1	nDNA	Increase the proportion of rubbed-out fibers (low complex II activity), Cox staining negative fibers (low complex IV activity), and Ragged Red Fibers.
Filaminopathy	FLNC	nDNA	Increase the proportion of Cox staining negative fibers (low complex IV activity), Ragged Red Fibers.
Myotilinopathy	MYOT	nDNA	Increase proportion of rubbed-out fibers (low complex II activity), Cox staining negative fibers (low complex IV activity)
Plectinopathy	PLEC	nDNA	Abnormal mitochondria distribution, increased proportion of rubbed-out fibers (low complex II activity), Cox staining negative fibers (low complex IV activity), and paracrystalline inclusions
Titinopathy	TTN	nDNA	Focal areas of mitochondrial depletion, increased proportion of Cox staining negative fibers (low complex IV activity), and paracrystalline inclusions
ZASPopathy	ZASP	nDNA	Increase proportion of rubbed-out fibers (low complex II activity), Cox staining negative fibers (low complex IV activity)

**Table 3 cells-12-02183-t003:** The major types of genetic mutation or deletion cause other myopathies with mitochondria phenotypes.

Myopathy	Gene of Mutation	nDNA/mtDNA	Muscle and Mitochondria Phenotypes	Reference
dermatomyositis	Unknown		Increase the proportion of SDH staining positive fiber and Cox staining negative fiber.	[188]
Pompe Disease	GAA	nDNA	Short and fragmented mitochondria reduce mitochondria respiration activity and ATP generation. Increase ROS generation. Mitochondria calcium overload. Increase expression of mitochondria dynamic-related protein but reduce mitophagy activity.	[189,192]
Metabolic myopathy	MIEF2	nDNA	Elongated mitochondria, aberrant mitochondrial cristae organization.	[190]
Dysferlinopathy	DYSF	nDNA	Decrease complex I, III, and IV protein level and activity, decrease cell ATP level.	[191]

**Table 4 cells-12-02183-t004:** Studies of aging-related alteration on mitochondrial dynamic and mitophagy.

Species	Sex	Tissue	Model	Protein Fraction	Mitochondria Dynamic Proteins	Mitophagy/Autophagy Related Proteins	Reference
Humans	Male	Vastus Lateralis	Younger men (20 ± 1 years) vs. older men (74 ± 3 years)	Whole muscle lysate	Mfn1, Mfn2 and Fis1: NS		[231]
Humans	Male	Vastus Lateralis	Younger men (22 ± 1 years) vs. older men (67 ± 2 years)	Whole muscle lysate	Opa1, Mfn2, Fis1: NS		[232]
Humans	Male and female (combined)	Vastus Lateralis	Younger (24 ± 3 years) vs. older adults (78 ± 5 years)	Whole muscle lysate	Opa1, Mfn2, Fis1 and Drp1: NS		[233]
Humans	Male and female (combined)	Vastus Lateralis	Younger (23 ± 1 years) vs. older adults (75 ± 1 years)	Whole muscle lysate	Mfn2, Fis1, Drp1: NS Opa1: ↓		[234]
Mice	Not specified	TA	Youths/Aged 6 vs. 18 months old	Whole muscle lysate	Opa1: ↓		[129]
Mice	Not specified	GAS	Youths/Aged 6 vs. 22 months old	Whole muscle lysate	Mfn1, Mfn2, Opa1 and Fis1: ↓ Drp1: NS	Lc3II, p62, Bnip3: ↑	[246]
Mice (C57BL/6J)	Female	TA	Youths/Aged 2 vs. 24 months old	Whole muscle lysate	Fis1 and Mfn2: ↑ Drp1 and Opa1: NS	Mitochondria Pink1, Parkin: ↑ Lc3I, Lc3II, p62, Rheb, Beclin1: ↑ Bnip3: ↓	[245]
Mice	Male	GAS	Youths/Aged 2–3 vs. 22–24 months old	Whole muscle lysate	Mfn2/Drp1 ratio: ↑ Opa1, Drp1, Mfn1 and Mfn2: NS		[244]
Mice (C57BL/6J)	Male	TA and soleus	Youths/Aged 3 vs. 28–29 months old	Whole muscle lysate and mitochondria fraction	Mfn2 (TA and SOL): ↑ Long Opa1 (functional)/Short Opa1 (nonfunctional) ratio (SOL): trend for ↑	Lc3II/Lc3I ratio (TA and SOL): NS Atg5 (TA and SOL): ↑ Mitochondria p62 (TA and SOL): ↑	[201]
Mice	Male and female (combined)	QUAD	Youths/Middle-aged 3–6 vs. 8–15 months old	Whole muscle lysate	Mfn1 and Mfn2: ↑ Opa1 and Drp1: NS Fis1: ↓	Beclin1: ↓ Ulk1: trend for ↓ p62: ↑ Lc3II, Atg5: NS	[243]
Rats (Fischer 344 Brown Norway)	Male	EDL	Youths/Aged 5 vs. 35 months old	Whole muscle lysate	Mfn2, Fis1 and Opa1: ↑ Drp1: NS	Ulk1, Beclin1, Atg7: ↑ Mitochondria Parkin, Lc3II: ↑	[236]
Rats (Wistar)	Not specified	GAS	Youths/Aged 3 vs. 26 months old	Mitochondria fraction	Fis1: ↑		[237]
Rats (Fischer 344 Brown Norway)	Male	TA	Youths/Aged 5 vs. 35 months old	Mitochondria fraction	Fis1 and Drp1: ↑ Mfn2: ↓ Opa1: NS		[238]
Rats (Sprague–Dawley)	Male	GAS and SOL	Youths/Aged 9 vs. 22 months old	Whole muscle lysate	Drp1 (SOL and GAS): ↑ Mfn2 and Fis1 (GAS): ↑		[239]
Rats (Wistar)	Male	GAS	Youths/Aged 3 vs. 26 months old	Whole muscle lysate	Fis1 and Mfn1: ↑		[240]
Rats (Sprague-Dawley)	Male	GAS and Triceps	Youths/Aged 3 vs. 22 months old	Whole muscle lysate	Opa1 and Mfn1 (GAS and TRI): ↑ Fis1 (GAS): ↓ Fis1 (TRI): ↑	Beclin1, Bax, Lc3B (GAS): ↓ Pink1 (Triceps): ↓	[241]
Rats (Sprague-Dawley)	Male	Muscles	Youths/Aged 5 vs. 25 months old	Whole muscle lysate	Drp1: ↑ Opa1: NS Mfn2 and Fis1: ↓	p62: ↑ Lc3II: ↓	[242]

## Data Availability

Not applicable.

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
