# Peer review of "Mitochondrial Properties in Skeletal Muscle Fiber"

_cells, 2023, doi:10.3390/cells12172183_

Round 1

Reviewer 1 Report

Dong and Tsai's review on mitochondria is extensive and comprehensive.  The authors do an adequate job of describing not only the molecular controls of mitochondria, but also the effect on human disease.  The authors cover the molecular control of mitochondrial biogenesis, oxidative stress, calcium homeostasis, and fission/fusion.  The authors also provide a summary of not only mitochondrial myopathies but also other myopathies with mitochondrial aspects.  Although the review can be viewed as a bit wordy in some areas, the authors writing is well done.  The one main recommendation from this reviewer is an incorporation of a figure or figures in the sections covering the molecular control of mitochondria.  Some readers may benefit from a visualization of how these molecules interact with each other to produce the effect. 

Author Response

Dong and Tsai's review on mitochondria is extensive and comprehensive.  The authors do an adequate job of describing not only the molecular controls of mitochondria, but also the effect on human disease.  The authors cover the molecular control of mitochondrial biogenesis, oxidative stress, calcium homeostasis, and fission/fusion.  The authors also provide a summary of not only mitochondrial myopathies but also other myopathies with mitochondrial aspects.  Although the review can be viewed as a bit wordy in some areas, the authors writing is well done.  The one main recommendation from this reviewer is an incorporation of a figure or figures in the sections covering the molecular control of mitochondria.  Some readers may benefit from a visualization of how these molecules interact with each other to produce the effect. 

Dear Reviewr,

Thank you so much for your kindly recommendation. In this revision, we add one figure to help the readers to visualize the control of mitochondria dynamic, and also show the link between the muscle fiber type specific mitochondria dynamic properties with the age-related muscle atrophy.

Hope this figure can help the readers to understand our review better.

Reviewer 2 Report

This is an excellent review article summarizing mitochondrial function in skeletal muscle.

However, only a small portion of the title describes muscle fiber-specific mitochondrial properties, and most of the rest describes mitochondrial function in skeletal muscle, not just muscle fibers.

I recommend that you change the title to something more appropriate for the text. I also think that the description in the summary section at the end should be more in line with the content of the main text.

Many abbreviations appear, but the full spelling is not always given where it first appears, please correct this.

Author Response

This is an excellent review article summarizing mitochondrial function in skeletal muscle.

However, only a small portion of the title describes muscle fiber-specific mitochondrial properties, and most of the rest describes mitochondrial function in skeletal muscle, not just muscle fibers.

I recommend that you change the title to something more appropriate for the text. I also think that the description in the summary section at the end should be more in line with the content of the main text.

Many abbreviations appear, but the full spelling is not always given where it first appears, please correct this.

Dear Reviewer,

Thank you so much for your kind recommendation. According to your comments, we have changed our title and conclusion part to avoid any misunderstood that may happen.

Reviewer 3 Report

In this manuscript, Dong et al. comprehensively summarized the role of skeletal muscle mitochondria in physiological and pathological conditions. The review is well-written and there are no major issues.

Minor points.

On page 4, I appreciate that the author summarized the role of muscle ERRalpha and gamma. Please also cite the following article, which was published at the same time as Sopariwala et al.

Wattez JS, Eury E, Hazen BC, Wade A, Chau S, Ou SC, Russell AP, Cho Y, Kralli A. Loss of skeletal muscle estrogen-related receptors leads to severe exercise intolerance. Mol Metab. 2023 Feb;68:101670. doi: 10.1016/j.molmet.2023.101670. Epub 2023 Jan 13. PMID: 36642217; PMCID: PMC9938320.

Author Response

In this manuscript, Dong et al. comprehensively summarized the role of skeletal muscle mitochondria in physiological and pathological conditions. The review is well-written and there are no major issues.

Minor points.

On page 4, I appreciate that the author summarized the role of muscle ERRalpha and gamma. Please also cite the following article, which was published at the same time as Sopariwala et al.

Wattez JS, Eury E, Hazen BC, Wade A, Chau S, Ou SC, Russell AP, Cho Y, Kralli A. Loss of skeletal muscle estrogen-related receptors leads to severe exercise intolerance. Mol Metab. 2023 Feb;68:101670. doi: 10.1016/j.molmet.2023.101670. Epub 2023 Jan 13. PMID: 36642217; PMCID: PMC9938320.

Dear Reviewer,

Thank you so much for your kind recommendation. The article you mentioned in the comments is very interesting and novel. In this revision, we have cited this paper in our main text and discuss about the unique findings of this excellent study.